# No Representation Rules Them All in Category Discovery

Sagar Vaze        Andrea Vedaldi        Andrew Zisserman

Visual Geometry Group
University of Oxford

www.robots.ox.ac.uk/~vgg/data/clevr4/

## Abstract

In this paper we tackle the problem of Generalized Category Discovery (GCD). Specifically, given a dataset with labelled and unlabelled images, the task is to cluster all images in the unlabelled subset, whether or not they belong to the labelled categories. Our first contribution is to recognize that most existing GCD benchmarks only contain labels for a single clustering of the data, making it difficult to ascertain whether models are using the available labels to solve the GCD task, or simply solving an unsupervised clustering problem. As such, we present a synthetic dataset, named 'Clevr-4', for category discovery. Clevr-4 contains four equally valid partitions of the data, *i.e.* based on object *shape*, *texture*, *color* or *count*. To solve the task, models are required to extrapolate the taxonomy specified by the labelled set, rather than simply latching onto a single natural grouping of the data. We use this dataset to demonstrate the limitations of unsupervised clustering in the GCD setting, showing that even very strong unsupervised models fail on Clevr-4. We further use Clevr-4 to examine the weaknesses of existing GCD algorithms, and propose a new method which addresses these shortcomings, leveraging consistent findings from the representation learning literature to do so. Our simple solution, which is based on 'mean teachers' and termed $\mu$GCD, substantially outperforms implemented baselines on Clevr-4. Finally, when we transfer these findings to real data on the challenging Semantic Shift Benchmark (SSB), we find that $\mu$GCD outperforms all prior work, setting a new state-of-the-art.

## 1   Introduction

Developing algorithms which can classify images within complex visual taxonomies, *i.e.* image recognition, remains a fundamental task in machine learning [1–3]. However, most models require these taxonomies to be *pre-defined* and *fully specified*, and are unable to construct them automatically from data. The ability to *build* a taxonomy is not only desirable in many applications, but is also considered a core aspect of human cognition [4–6]. The task of constructing a taxonomy is epitomized by the Generalized Category Discovery (GCD) problem [7, 8]: given a dataset of images which is labelled only in part, the goal is to label all remaining images, using categories that occur in the labelled subset, or by identifying new ones. For instance, in a supermarket, given only labels for 'spaghetti' and 'penne' pasta products, a model must understand the concept of 'pasta shape' well enough to generalize to 'macaroni' and 'fusilli'. It must *not* cluster new images based on, for instance, the color of the packaging, even though the latter *also* yields a valid, but different, taxonomy.

GCD is related to self-supervised learning [9] and unsupervised clustering [10], which can discover *some* meaningful taxonomies automatically [11]. However, these *cannot* solve the GCD problem, which requires recovering *any* of the different and incompatible taxonomies that apply to the same data. Instead, the key to GCD is in *extrapolating a taxonomy* which is only partially known. In this paper, our objective is to better understand the GCD problem and improve algorithms' performance.

37th Conference on Neural Information Processing Systems (NeurIPS 2023).

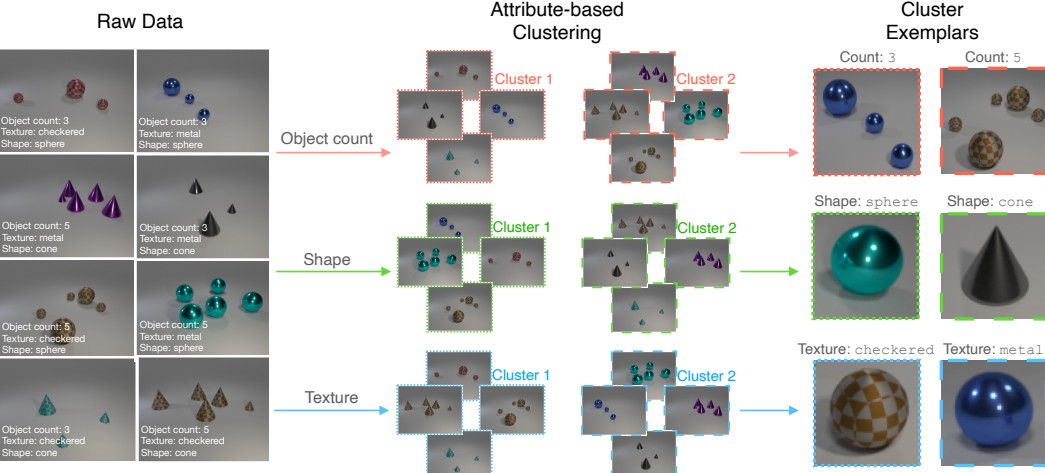

Figure 1: What is the key difference between Generalized Category Discovery (GCD) and tasks like self-supervised learning or unsupervised clustering? GCD's key challenge is extrapolating the *desired* clustering of data given only a subset of possible category labels. We present a synthetic dataset, Clevr-4, which contains *four* possible clusterings of the same images, and hence can be used to isolate the GCD task. Above, one can cluster the data based on object *count*, *shape* or *texture*.

To this end, in section 3, we first introduce the Clevr-4 dataset. Clevr-4 is a synthetic dataset where each image is fully parameterized by a set of four attributes, and where each attribute defines an equally valid grouping of the data (see fig. 1). Clevr-4 extends the original CLEVR dataset [12] by introducing new *shapes*, *colors* and *textures*, as well as allowing different object *counts* to be present in the image. Using these four attributes, the same set of images can be clustered according to *four* statistically independent taxonomies. This feature sets it apart from most existing GCD benchmarks, which only contain sufficient annotations to evaluate a *single* clustering of the data.

Clevr-4 allows us to probe large pre-trained models for biases, *i.e.*, for their preference to emphasize a particular aspect of images, such as color or texture, which influences which taxonomy can be learned. For instance, contrary to findings from Geirhos *et al.* [13], we find almost every large model exhibits a strong shape bias. Specifically, in section 4, we find unsupervised clustering – even with very strong representations like DINO [14] and CLIP [15] – fails on many splits of Clevr-4, despite CLEVR being considered a 'toy' problem in other contexts [16]. As a result, we find that different pre-trained models yield different performance traits across Clevr-4 when used as initialization for category discovery. We further use Clevr-4 to characterize the weaknesses of existing category discovery methods; namely, the harms of jointly training feature-space and classifier losses, as well as insufficiently robust pseudo-labelling strategies for 'New' classes.

Next, in section 5, we leverage our findings on Clevr-4 to develop a simple but performant method for GCD. Since category discovery has substantial overlap with self-supervised learning, we identify elements of these methods that are also beneficial for GCD. In particular, mean-teacher based algorithms [17] have been very effective in representation learning [14, 18], and we show that they can boost GCD performance as well. Here, a 'teacher' model provides supervision through pseudo-labels, and is maintained as the moving average of the model being trained (the 'student'). The slowly updated teacher is less affected by the noisy pseudo-labels which it produces, allowing clean pseudo-labels to be produced for new categories. Our proposed method, '$\mu$GCD' ('mean-GCD'), extends the existing state-of-the-art [19], substantially outperforming it on Clevr-4. Finally, in section 6, we compare $\mu$GCD against prior work on real images, by evaluating on the challenging Semantic Shift Benchmark (SSB) [20]. We substantially improve state-of-the-art on this evaluation.

In summary, we make the following key contributions: **(i)** We propose a new benchmark dataset, Clevr-4, for GCD. Clevr-4 contains four independent taxonomies and can be used to better study the category discovery problem. **(ii)** We use Clevr-4 to garner insights on the biases of large pre-trained models as well as the weaknesses of existing category discovery methods. We demonstrate that even very strong unsupervised models fail on this 'toy' benchmark. **(iii)** We present a novel method for GCD, $\mu$GCD, inspired by the mean-teacher algorithm. **(iv)** We show $\mu$GCD outperforms baselines on Clevr-4, and further sets a new state-of-the-art on the challenging Semantic Shift Benchmark.

## 2 Related Work

**Representation learning.** The common goal of self-, semi- and unsupervised learning is to learn representations with minimal labelled data. A popular technique is contrastive learning [21, 22], which encourages representations of different augmentations of the same training sample to be similar. Contrastive methods are typically either based on: InfoNCE [23] (*e.g.*, MoCo [24] and SimCLR [21]); or online pseudo-labelling (*e.g.*, SWaV [9] and DINO [14]). Almost all contrastive learning methods now adopt a variant of these techniques [25–28]. Another important component in many pseudo-labelling based methods is 'mean-teachers' [17] (or momentum encoders [24]), in which a 'teacher' network providing pseudo-labels is maintained as the moving average of a 'student' model. Other learning methods include cross-stitch [29], context-prediction [30], and reconstruction [31]. In this work, we use mean-teachers to build a strong recipe for GCD.

**Attribute learning.** We propose a new synthetic dataset which contains multiple taxonomies based on various attributes. Attribute learning has a long history in computer vision, including real-world datasets such as the Visual Genome [32], with millions of attribute annotations, and VAW, with 600 attributes types [33]. Furthermore, the *disentanglement literature* [34–36] often uses synthetic attribute datasets for investigation [37, 38]. We find it necessary to develop a new dataset, Clevr-4, for category discovery as real-world datasets have either: noisy/incomplete attributes for each image [32, 39]; or contain sensitive information (*e.g.* contain faces) [40]. We find existing synthetic datasets unsuitable as they do not have enough categorical attributes which represent 'semantic' factors, with attributes often describing continuous 'nuisance' factors such as object location or camera pose [37, 38, 41].

**Category Discovery.** Novel Category Discovery (NCD) was initially formalized in [42]. It differs from GCD as the unlabelled images are known to be drawn from a *disjoint* set of categories to the labelled ones [43–47]. This is different from *unsupervised clustering* [10, 48], which clusters unlabelled data without reference to labels at all. It is also distinct from *semi-supervised learning* [17, 26, 49], where unlabelled images come from the *same* set of categories as the labelled data. GCD [7, 8] was recently proposed as a challenging task in which assumptions about the classes in the unlabelled data are largely removed: images in the unlabelled data may belong to the labelled classes or to new ones [19, 50–52]. We particularly highlight concurrent work in SimGCD [19], which reports the best current performance on standard GCD benchmarks. Our method differs from SimGCD by the adoption of a mean-teacher [17] to provide more stable pseudo-labels training, and by careful consideration of model initialization and data augmentations. [52] also adopt a momentum-encoder, though only for a set of class prototypes rather than in a mean-teacher setup.

## 3 Clevr-4: a synthetic dataset for generalized category discovery

**Generalized Category Discovery (GCD).** GCD [7] is the task of fully labelling a dataset which is only partially labelled, with labels available only for a subset of the categories. Formally, we are presented with a dataset $\mathcal{D}$, containing both labelled and unlabelled subsets, $\mathcal{D}_{\mathcal{L}} = \{(\boldsymbol{x}_i, y_i)\}_{i=1}^{N_l} \in \mathcal{X} \times \mathcal{Y}_{\mathcal{L}}$ and $\mathcal{D}_{\mathcal{U}} = \{(\boldsymbol{x}_i, y_i)\}_{i=1}^{N_u} \in \mathcal{X} \times \mathcal{Y}_{\mathcal{U}}$. The model is then tasked with categorizing all instances in $\mathcal{D}_{\mathcal{U}}$. The unlabelled data contains instances of categories which do not appear in the labelled set, meaning that $\mathcal{Y}_{\mathcal{L}} \subset \mathcal{Y}_{\mathcal{U}}$. Here, we *assume knowledge* of the number of categories in the unlabelled set, *i.e.* $k = |\mathcal{Y}_{\mathcal{U}}|$. Though estimating $k$ is an interesting problem, methods for tackling it are typically independent of the downstream recognition algorithm [7, 42]. GCD is related to Novel Category Discovery (NCD) [42], but the latter makes the not-so-realistic assumption that $\mathcal{Y}_{\mathcal{L}} \cap \mathcal{Y}_{\mathcal{U}} = \emptyset$.

**The category discovery challenge.** The key to category discovery (generalized or not) is to use the labelled subset of the data to *extrapolate a taxonomy* and discover novel categories in unlabelled images. This task of extrapolating a taxonomy sets category discovery apart from related problems. For instance, unsupervised clustering [10] aims to find the single most natural grouping of unlabelled images given only weak inductive biases (*e.g.*, invariance to specific data augmentations), but permits limited control on *which* taxonomy is discovered. Meanwhile, semi-supervised learning [26] assumes supervision for all categories in the taxonomy, which therefore must be known, in full, a-priori.

A problem with many current benchmarks for category discovery is that there is no clear taxonomy underlying the object categories (*e.g.*, CIFAR [53]) and, when there is, it is often ill-posed to understand it given only a few classes (*e.g.*, ImageNet-100 [42]). Furthermore, in practise, there

are likely to be many taxonomies of interest. However, few datasets contain sufficiently complete annotations to evaluate multiple possible groupings of the same data. This makes it difficult to ascertain whether a model is extrapolating information from the labelled set (category discovery) or just finding its own most natural grouping of the unlabelled data (unsupervised clustering).

**Clevr-4.** In order to better study this problem, we introduce Clevr-4, a synthetic benchmark which contains four equally valid groupings of the data. Clevr-4 extends the CLEVR dataset [12], using Blender [54] to render images of multiple objects and place them in a static scene. This is well suited for category discovery,

Table 1: **Clevr-4 statistics** for the different splits of the dataset. Note that the *same data* must be classified along independent taxonomies in the different splits.

|  | Texture | Color | Shape | Count |
|---|---|---|---|---|
| Examples | {metal,rubber} | {red,blue} | {torus,cube} | {1,2} |
| $|\mathcal{Y}_{\mathcal{L}}|$ | 5 | 5 | 5 | 5 |
| $|\mathcal{Y}_{\mathcal{U}}|$ | 10 | 10 | 10 | 10 |
| $|\mathcal{D}_{\mathcal{L}}|$ | 2.1K | 2.3K | 2.1K | 2.1K |
| $|\mathcal{D}_{\mathcal{U}}|$ | 6.3K | 6.1K | 6.4K | 6.3K |
| $|\mathcal{D}_{\mathcal{L}}| + |\mathcal{D}_{\mathcal{U}}|$ | 8.4K | 8.4K | 8.4K | 8.4K |

as each object attribute defines a different taxonomy for the data (*e.g.*, it enables clustering images based on object *shape*, *color* etc.). The original dataset is limited as it contains only three shapes and two textures, reducing the difficulty of the respective clustering tasks. We introduce 2 new colors, 7 new shapes and 8 new textures to the dataset, placing between 1 and 10 objects in each scene.

Each image is therefore parameterized by object *shape*, *texture*, *color* and *count*. The value for each attribute is sampled uniformly and independently from the others, meaning the image label with respect to one taxonomy gives us no information about the label with respect to another. Note that this sets Clevr-4 apart from existing GCD benchmarks such as CIFAR-100 [53] and FGVC-Aircraft [55]. These datasets only contain taxonomies at different *granularities*, and as such the taxonomies are highly correlated with each other. Furthermore, the number of categories provides no information regarding the specified taxonomy, as all Clevr-4 taxonomies contain $k = 10$ object categories.

Finally, we create GCD splits for each taxonomy in Clevr-4, following standard practise and reserving half the categories for the labelled set, and half for the unlabelled set. We further subsample 50% of the images from the labelled categories and add them to the unlabelled set. We synthesize $8.4K$ images for GCD development (summarized in table 1), and further make a larger $100K$ image dataset available. The full generation procedure is detailed in appendix A.1.

**Performance metrics.** We follow standard practise [7, 47] and report *clustering accuracy* for evaluation. Given predictions, $\hat{y}_i$, and ground truth labels, $y_i$, the clustering accuracy is $ACC = \max_{\Pi \in S_k} \frac{1}{N_u} \sum_{i=1}^{N_u} \mathbb{1}\{\hat{y}_i = \Pi(y_i)\}$, where $S_k$ is the symmetry group of order $k$. Here $N_u$ is the number of unlabelled images and the max operation is performed (with the Hungarian algorithm [56]) to find the optimal matching between the predicted cluster indices and ground truth labels. For GCD models, we also report $ACC$ on subsets belonging to 'Old' ($y_i \in \mathcal{Y}_{\mathcal{L}}$) and 'New' ($y_i \in \mathcal{Y}_{\mathcal{U}} \setminus \mathcal{Y}_{\mathcal{L}}$) classes. The most important metric is the 'All' accuracy (overall clustering performance), as the precise 'Old' and 'New' figures are subject to the assignments selected in the Hungarian matching.

## 4 Learnings from Clevr-4 for category discovery

In this section, we use Clevr-4 to gather insights into the category discovery problem. In section 4.1, we assess the limitations of large-scale pre-training for the problem, before using Clevr-4 to examine the weaknesses of existing category discovery methods in section 4.2. Next, in section 4.3, we use our findings to motivate a stronger method for GCD (which we describe in full in section 5), finding that this substantially outperforms implemented baselines on Clevr-4.

### 4.1 Limitations of large-scale pre-training for category discovery

Here, we show that pre-trained representations develop certain 'biases' which limit their performance when used directly or as initialization for category discovery.

**Unsupervised clustering of pre-trained representations (table 2).** We first demonstrate the limitations of unsupervised clustering of features as an approach for category discovery (reporting results with semi-supervised clustering in fig. 11). Specifically, we run $k$-means clustering [57] on top of features extracted with self- [9, 14, 21], weakly- [15], and fully-supervised [2, 3, 58] backbones, reporting performance on each of the four taxonomies in Clevr-4. The representations are trained on up to 400M images and are commonly used in the vision literature.

Table 2: **Unsupervised clustering accuracy (ACC) of pre-trained models on Clevr-4.** We find most models are strongly biased towards *shape*, while MAE [31] exhibits a *color* bias.

| Pre-training Method | Pre-training Data | Backbone | Texture | Shape | Color | Count | Average |
|---|---|---|---|---|---|---|---|
| SWaV [9] | ImageNet-1K | ResNet50 | 13.1 | 65.5 | 12.1 | **18.9** | 27.4 |
| MoCoV2 [22] | ImageNet-1K | ResNet50 | 13.0 | 77.5 | 12.3 | 18.8 | 30.4 |
| Supervised [2] | ImageNet-1K | ResNet50 | 13.2 | 76.8 | 15.2 | 12.9 | 29.5 |
| Supervised [3] | ImageNet-1K | ConvNeXT-B | 13.4 | 83.5 | 12.1 | 13.1 | 30.5 |
| DINO [14] | ImageNet-1K | ViT-B/16 | **16.0** | 86.2 | 11.5 | 13.0 | 31.7 |
| MAE [31] | ImageNet-1K | ViT-B/16 | 15.1 | 13.5 | **64.7** | 13.9 | 26.8 |
| iBOT [59] | ImageNet-1K | ViT-B/16 | 14.4 | 85.9 | 11.5 | 13.0 | 31.2 |
| CLIP [15] | WIP-400M | ViT-B/16 | 12.4 | 78.7 | 12.3 | 17.9 | 30.3 |
| DINOv2 [60] | LVD-142M | ViT-B/14 | 11.6 | **98.1** | 11.6 | 12.8 | **33.5** |
| Supervised [58] | ImageNet-21K | ViT-B/16 | 11.8 | 96.2 | 11.7 | 13.0 | 33.2 |

Table 3: **Effects of large-scale pre-training on category discovery accuracy (ACC) on Clevr-4.** Contrary to dominant findings in the vision literature, we find that large-scale pre-training provides inconsistent gains on Clevr-4. For instance, on *count*, training a lightweight model (ResNet18) from scratch substantially outperforms initializing from DINOv2 (ViT-B/14) trained on 142M images.

| Method | Backbone | Pre-training (Data) | Texture | Shape | Color | Count | Average | Average Rank |
|---|---|---|---|---|---|---|---|---|
| SimGCD | ResNet18 | - | 58.1 | 97.8 | 96.7 | **67.6** | **80.5** | **2.0** |
| SimGCD | ViT-B/16 | MAE [31] (ImageNet-1k) | 54.1 | 99.7 | **99.9** | 53.0 | 76.7 | **2.0** |
| SimGCD | ViT-B/14 | DINOv2 [60] (LVD-142M) | **76.5** | **99.9** | 87.4 | 51.3 | 78.8 | **2.0** |

We find that most models perform well on the *shape* taxonomy, with DINOv2 almost perfectly solving the task with 98% accuracy. However, none of the models perform well across the board. For instance, on some splits (*e.g.*, *color*), strong models like DINOv2 perform comparably to random chance. This underscores the utility of Clevr-4 for delineating category discovery from standard representation learning. Logically, it is *impossible* for unsupervised clustering on *any* representation to perform well on all tasks. After all, only a single clustering of the data is produced, which cannot align with more than one taxonomy. We highlight that such limitations are *not* revealed by existing GCD benchmarks; on the CUB benchmark (see table 5), unsupervised clustering with DINOv2 achieves 68% ACC ($\approx 140\times$ random).

**Pre-trained representations for category discovery (table 3).** Many category discovery methods use self-supervised representation learning for initialization in order to leverage large-scale pre-training, in the hope of improving downstream performance. However, as shown above, these representations are biased. Here, we investigate the impact of these biases on a state-of-the-art method in generalized category discovery, SimGCD [19]. SimGCD contains two main loss components: (1) a contrastive loss on backbone features, using self-supervised InfoNCE [23] on all data, and supervised contrastive learning [61] on images with labels available; and (2) a contrastive loss to train a classification head, where different views of the same image provide pseudo-labels for each other. For comparison, we initialize SimGCD with a lightweight ResNet18 trained scratch; a ViT-B/16 pre-trained with masked auto-encoding [31]; and a ViT-B/14 with DINOv2 [60] initialization. For each initialization, we sweep learning rates and data augmentations.

Surprisingly, and in stark contrast to most of the computer vision literature, we find inconsistent gains from leveraging large-scale pre-training on Clevr-4. For instance, on the *count* taxonomy, pre-training gives substantially *worse* performance that training a lightweight ResNet18 from scratch. On average across all splits, SimGCD with a randomly initialized ResNet18 actually performs best. Generally, we find that the final category discovery model inherits biases built into the pre-training, and can struggle to overcome them even after finetuning. Our results highlight the importance of carefully selecting the initialization for a given GCD task, and point to the utility of Clevr-4 for doing so.

## 4.2 Limitations of existing category discovery methods

Next, we analyze SimGCD [19], the current state-of-the-art for the GCD task. We show that on Clevr-4 it is not always better than the GCD baseline [7] which it extends, and identify the source of this issue in the generation of the pseudo-labels for the discovered categories. In more detail, the

Table 4: **Category discovery accuracy (ACC) on Clevr-4**. Compared to our reimplementation of the GCD baseline [7] and SimGCD state-of-the-art [19], our method provides substantial boosts on average across Clevr-4. Results are averages across five random seeds. Also shown is the classification accuracy of a fully-supervised upper-bound on a disjoint test-set.

| Model | Backbone | Texture | | | Shape | | | Color | | | Count | | | Average |
|---|---|---|---|---|---|---|---|---|---|---|---|---|---|---|
| | | All | Old | New | All | Old | New | All | Old | New | All | Old | New | All |
| Fully supervised | ResNet18 | 99.1 | - | - | 100.0 | - | - | 100.0 | - | - | 96.8 | - | - | 99.0 |
| GCD | ResNet18 | 62.4 | 97.5 | 45.3 | 93.9 | 99.7 | 90.5 | 90.7 | 95.0 | 88.5 | 71.9 | 96.4 | 60.1 | 79.7 |
| SimGCD | ResNet18 | 58.1 | 95.0 | 40.2 | **97.8** | 98.9 | **97.2** | 96.7 | 99.9 | 95.1 | 67.6 | 95.7 | 53.9 | 80.1 |
| $\mu$GCD (Ours) | ResNet18 | **69.8** | **99.0** | **55.5** | 94.9 | 99.7 | 92.1 | **99.5** | **100.0** | **99.2** | **75.5** | 96.6 | **65.2** | **84.9** |

GCD baseline uses only one of the two losses used by SimGCD, performing contrastive learning on features, followed by simple clustering in the models' embedding space. To compare SimGCD and GCD, we start from a ResNet18 feature extractor, training it from scratch to avoid the potential biases identified in section 4.1. We show results in table 4, reporting results for 'All', 'Old' and 'New' class subsets. We follow standard practise when reporting on synthetic data and train all methods with five random seeds [34, 35] (error bars in the appendix B.1), and sweep hyper-parameters and data augmentations for both methods. We also train a model with full supervision and obtain 99% average performance on Clevr-4 (on independent test data), showing that the model has sufficient capacity.

Overall, we make the three following observations regarding the performance of GCD and SimGCD on Clevr-4: **(i)** Both methods' performance on *texture* and *count* is substantially worse than on *shape* and *color*. **(ii)** On the harder *texture* and *count* splits, the GCD baseline actually outperforms the SimGCD state-of-the-art. Given that SimGCD differs from GCD by adding a classification head and corresponding loss, this indicates that jointly training classifier and feature-space losses can hurt performance. **(iii)** Upon closer inspection, we find that the main performance gap on *texture* and *count* comes from accuracy on the 'New' categories; both methods cluster the 'Old' categories almost perfectly. This suggests that the 'New' class pseudo-labels from SimGCD are not strong enough; GCD, with no (pseudo-)supervision for novel classes, achieves higher clustering performance.

### 4.3   Addressing limitations in current approaches

Given these findings, we seek to improve the quality of the pseudo-labels for 'New' categories. Specifically, we draw inspiration from the mean-teacher setup for semi-supervised learning [17], which has been adapted with minor changes in many self-supervised frameworks [14, 18, 24]. Here, a 'student' network is supervised by class pseudo-labels generated by a 'teacher'. The teacher is an identical architecture with parameters updated with the Exponential Moving Average (EMA) of the student. The intuition is that the slowly updated teacher is more robust to the noisy supervision from pseudo-labels, which in turn improves the quality of the pseudo-labels themselves. Also, rather than *jointly optimizing* both SimGCD losses, we first train the backbone *only* with the GCD baseline loss, before *finetuning* with the classification head and loss.

These changes, together with careful consideration of the data augmentations, give rise to our proposed $\mu$GCD (mean-GCD) algorithm, which we fully describe next in section 5. Here, we note the improvements that this algorithm brings in Clevr-4 on the bottom line of table 4. Overall, $\mu$GCD outperforms SimGCD on three of the four Clevr-4 taxonomies, and further outperforms SimGCD by nearly 5% on average across all splits. $\mu$GCD underperforms SimGCD on the *shape* split of Clevr-4 and we analyse this failure case in the appendix B.2.

## 5   The $\mu$GCD algorithm

In this section, we detail a simple but strong method for GCD, $\mu$GCD, already motivated in section 4 and illustrated in fig. 2. In a first phase, the algorithm proceeds in the same way as the GCD baseline [7], learning the representation. Next, we append a classification head and fine-tune the model with a 'mean teacher' setup [17], similarly to SimGCD but yielding more robust pseudo-labels.

Concretely, we construct models, $f_\theta$, as the composition of a feature extractor, $\Phi$, and a classification head, $g$. $\Phi$ is first trained with the representation learning framework from [7] as described above, and the composed model gives $f = g \circ \Phi$ with values in $\mathbb{R}^k$, where $k$ is the total number of categories in the dataset. Next, we sample a batch of images, $\mathcal{B}$, and generate two random augmentations of

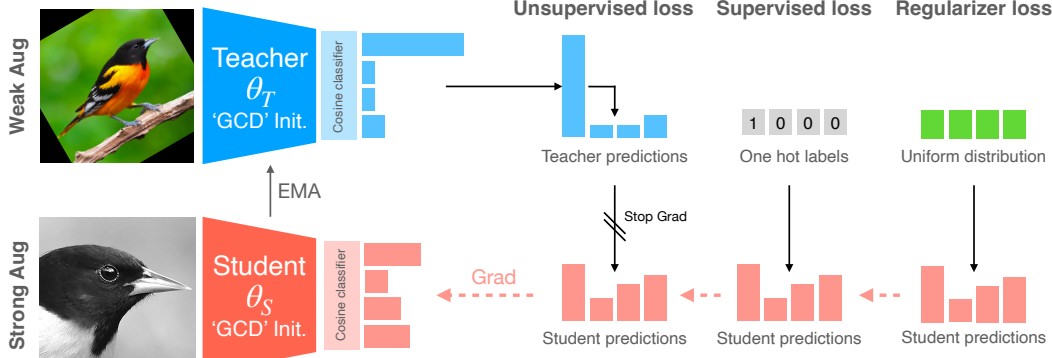

Figure 2: **Our 'μGCD' method**. We begin with representation learning from the GCD baseline, followed by finetuning in a mean-teacher style setup. Here, a 'teacher' provides supervision for a 'student' network, and maintains parameters as the exponential moving average (EMA) of the student.

every instance. We pass one view through the student network $f_{\theta_S}$, and the other through the teacher network $f_{\theta_T}$, where $\theta_S$ and $\theta_T$ are the network parameters of the student and teacher, respectively. We compute the cross-entropy loss between the (soft) teacher pseudo-labels and student predictions:

$$\mathcal{L}^u(\theta_S; \mathcal{B}) = -\frac{1}{|\mathcal{B}|} \sum_{\boldsymbol{x} \in \mathcal{B}} \langle \boldsymbol{p}_T(\boldsymbol{x}), \log(\boldsymbol{p}_S(\boldsymbol{x})) \rangle, \quad \boldsymbol{p}_*(\boldsymbol{x}) = \mathrm{softmax}(f_{\theta_*}(\boldsymbol{x}); \tau_*), \quad (1)$$

where $\boldsymbol{p}_*(\boldsymbol{x}) \in [0,1]^k$ are the softmax outputs of the student and teacher networks, scaled with temperature $\tau_*$. We further use labelled instances in the batch with a supervised cross-entropy component as:

$$\mathcal{L}^s(\theta_S; \mathcal{B}_\mathcal{L}) = -\frac{1}{|\mathcal{B}_\mathcal{L}|} \sum_{i \in \mathcal{B}_\mathcal{L}} \langle \boldsymbol{y}(\boldsymbol{x}), \log(\boldsymbol{p}_S(\boldsymbol{x})) \rangle, \quad (2)$$

where $\mathcal{B}_\mathcal{L} \in \mathcal{B}$ is the labelled subset of the batch and $\boldsymbol{y}(\boldsymbol{x}) \in \{0,1\}^k$ is the one-hot class label of the example $\boldsymbol{x}$. Finally, we add a mean-entropy maximization regularizer from [26] to encourage pseudo-labels for all categories:

$$\mathcal{L}^r(\theta_S) = -\langle \bar{\boldsymbol{p}}_S, \log(\bar{\boldsymbol{p}}_S) \rangle, \qquad \bar{\boldsymbol{p}}_S = \frac{1}{|\mathcal{B}|} \sum_{\boldsymbol{x} \in \mathcal{B}} \boldsymbol{p}_S(\boldsymbol{x}). \quad (3)$$

The student is trained with respect to the following total loss, given hyper-parameters $\lambda_1$ and $\lambda_2$: $\mathcal{L}(\theta_S; \mathcal{B}) = (1 - \lambda_1)\mathcal{L}^u(\theta_S; \mathcal{B}) + \lambda_1 \mathcal{L}^s(\theta_S; \mathcal{B}_\mathcal{L}) + \lambda_2 \mathcal{L}^r(\theta_S)$. The teacher parameters are updated as the moving average $\theta_T = \omega(t)\theta_T + (1 - \omega(t))\theta_S$, where $\omega(t)$ is a time-varying momentum.

**Augmentations.** While often regarded as an 'implementation detail', an important component of our method is the careful consideration of augmentations used in the computation of $\mathcal{L}^u$. Specifically, on the SSB, we pass different views of the same instance to the student and teacher networks. We generate a *strong* augmentation which is passed to the student network, and a *weak* augmentation which is passed to the teacher, similarly to [49]. The intuition is that, while contrastive learning benefits from strong data augmentations [9, 14], we wish the teacher network's predictions to be as stable as possible. Meanwhile, on Clevr-4, misaligned data augmentations — *e.g.*, aggressive cropping for *count*, or color jitter for *color* — substantially degrade performance (see appendix B.5).

**Architecture.** We adopt a 'cosine classifier' as $g$, which was introduced in [62] and leverages $L^2$-normalized weight vectors and feature representations. While it has been broadly adopted for many tasks [8, 9, 19, 25, 43], we demonstrate *why* this component helps in section 7.1. We find that normalized vectors are important to avoid collapse of the predictions to the labelled categories.

## 6 Results on real data

**Datasets.** We compare μGCD against prior work on the standard Semantic Shift Benchmark (SSB) suite [20]. The SSB comprises three fine-grained evaluations: CUB [64], Stanford Cars [65] and

Table 5: **Category discovery accuracy (ACC) on the Semantic Shift Benchmark [20].** We report results from prior work using DINO intialization [14], and reimplement GCD baselines and SimGCD with DINOv2 pre-training [60] (noted with *).

| | Pre-training | CUB | | | Stanford Cars | | | Aircraft | | | Average |
|---|---|---|---|---|---|---|---|---|---|---|---|
| | | All | Old | New | All | Old | New | All | Old | New | All |
| $k$-means [57] | DINO | 34.3 | 38.9 | 32.1 | 12.8 | 10.6 | 13.8 | 16.0 | 14.4 | 16.8 | 21.1 |
| RankStats+ [47] | DINO | 33.3 | 51.6 | 24.2 | 28.3 | 61.8 | 12.1 | 26.9 | 36.4 | 22.2 | 29.5 |
| UNO+ [43] | DINO | 35.1 | 49.0 | 28.1 | 35.5 | 70.5 | 18.6 | 40.3 | 56.4 | 32.2 | 37.0 |
| ORCA [8] | DINO | 35.3 | 45.6 | 30.2 | 23.5 | 50.1 | 10.7 | 22.0 | 31.8 | 17.1 | 26.9 |
| GCD [7] | DINO | 51.3 | 56.6 | 48.7 | 39.0 | 57.6 | 29.9 | 45.0 | 41.1 | 46.9 | 45.1 |
| XCon [63] | DINO | 52.1 | 54.3 | 51.0 | 40.5 | 58.8 | 31.7 | 47.7 | 44.4 | 49.4 | 46.8 |
| OpenCon [52] | DINO | 54.7 | 63.8 | 54.7 | 49.1 | 78.6 | 32.7 | - | - | - | - |
| MIB [51] | DINO | 62.7 | 75.7 | 56.2 | 43.1 | 66.9 | 31.6 | - | - | - | - |
| PromptCAL [50] | DINO | 62.9 | 64.4 | 62.1 | 50.2 | 70.1 | 40.6 | 52.2 | 52.2 | 52.3 | 55.1 |
| SimGCD [19] | DINO | 60.3 | 65.6 | 57.7 | 53.8 | 71.9 | 45.0 | 54.2 | 59.1 | 51.8 | 56.1 |
| $\mu$GCD (Ours) | DINO | 65.7 | 68.0 | 64.6 | 56.5 | 68.1 | 50.9 | 53.8 | 55.4 | 53.0 | 58.7 |
| $k$-means* | DINOv2 | 67.6 | 60.6 | 71.1 | 29.4 | 24.5 | 31.8 | 18.9 | 16.9 | 19.9 | 38.6 |
| GCD* | DINOv2 | 71.9 | 71.2 | 72.3 | 65.7 | 67.8 | 64.7 | 55.4 | 47.9 | 59.2 | 64.3 |
| SimGCD* | DINOv2 | 71.5 | **78.1** | 68.3 | 71.5 | 81.9 | 66.6 | 63.9 | **69.9** | 60.9 | 69.0 |
| $\mu$GCD (Ours) | DINOv2 | **74.0** | 75.9 | **73.1** | **76.1** | **91.0** | **68.9** | **66.3** | 68.7 | **65.1** | **72.1** |

FGVC-Aircraft [55]. Though the SSB datasets do not contain independent clusterings of the same images (as in Clevr-4) the evaluations do have well-defined taxonomies — *i.e.* birds, cars and aircrafts. Furthermore, the SSB contains curated novel class splits which control for semantic distance with the labelled set. We find that coarse-grained GCD benchmarks do not specify clear taxonomies in the labelled set, and we include a long-tailed evaluation on Herbarium19 [66] in appendix C.3.

**Model initialization and compared methods.** The SSB contains fine-grained, object-centric datasets, which have been shown to benefit from greater shape bias [67]. Prior GCD methods [7, 52, 63] initialize with DINO [14] pre-training, which we show in table 2 had the strongest shape bias among self-supervised models. However, the recent DINOv2 [60] demonstrates a substantially greater shape bias. As such, we train our model both with DINO and DINOv2 initialization, further re-implementing GCD baselines [7, 68] and SimGCD [19] with DINOv2 for comparison.

Table 6: **Ablations.** We find that a proper intialization, momentum decay schedule, and augmentation strategy are critical to strong performance.

| | CUB | | |
|---|---|---|---|
| | All | Old | New |
| $\mu$GCD (Ours) | **65.7** | 68.0 | **64.6** |
| (1) W/o GCD init. | 61.7 | 66.2 | 59.6 |
| (2) W/o stronger student augmentation | 58.1 | **72.5** | 50.9 |
| (3) With $\omega_t := 1$ | 1.6 | 1.1 | 1.8 |
| (4) With $\omega_t := 0.0$ | 62.7 | 66.4 | 60.9 |
| (5) With $\omega_t := 0.7$ | 64.1 | 65.1 | 63.6 |
| (6) W/o cosine classifier | 54.9 | 64.2 | 50.3 |
| (7) W/o ME-Max regularizer | 42.0 | 41.8 | 42.1 |

**Implementation details.** We implement all models in PyTorch [69] on a single NVIDIA P40 or M40. Most models are trained with an initial learning rate of 0.1 which is decayed with a cosine annealed schedule [70]. For our EMA schedule, we ramp it up throughout training with a cosine function [18]: $\omega(t) = \omega_T - (1 - \omega_{base})(\cos(\frac{\pi t}{T}) + 1)/2$. Here $t$ is the current epoch and $T$ is the total number of epochs. Differently, however, to most self-supervised learning literature [18], we found a much lower initial decay to be beneficial; we ramp up the decay from $\omega_{base} = 0.7$ to $\omega_T = 0.999$ during training. Further implementation details can be found in appendix E.

## 6.1 Discussion.

In table 5, we find that $\mu$GCD outperforms the existing state-of-the-art, SimGCD [19], by over 2% on average across all SSB evaluations when using DINO initialization. When using the stronger DINOv2 backbone, we find that the performance of the simple $k$-means baseline nearly doubles in accuracy, substantiating our choice of shape-biased initialization on this object-centric evaluation. The gap between the GCD baseline [7] and the SimGCD state-of-the-art [19] is also reduced from over 10% to under 5% on average. Nonetheless, our method outperforms SimGCD by over 3% on average, as well as on each dataset individually, setting a new state-of-the-art.

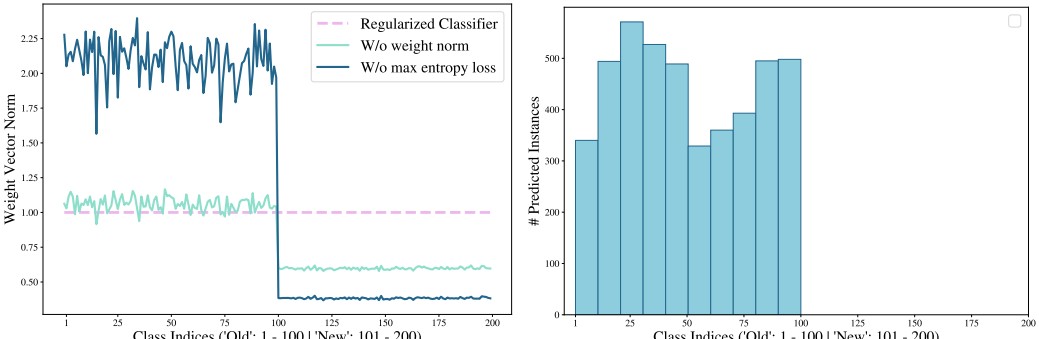

Figure 3: **Left:** Norms of weight vectors in GCD classifiers, with and without regularizations. **Right:** Prediction histogram of a classifier without weight norm and max-entropy regularization.

**Ablations.** We ablate our main design choices in table 6. L(1) shows the importance of pre-training with the GCD baseline loss [7] (though we find in section 4 that jointly training this loss with the classifier, as in SimGCD [19], is difficult). L(2) further demonstrates that stronger augmentation for the student network is critical, with a 7% drop in CUB performance without it. L(3)-(5) highlight the importance of a carefully designed EMA schedule, our use of a time-varying decay outperforms constant decay values. This is intuitive as early on in training, with a randomly initialized classification head, we wish for the teacher to be updated quickly. Later on in training, slow teacher updates mitigate the effect of noisy pseudo-labels within any given batch. Furthermore, in L(6)-(7), we validate the importance of entropy regularization and cosine classifiers in category discovery. In section 7.1, we provide evidence as to *why* these commonly used components [8, 19, 43] are necessary, and also discuss the design of the student augmentation.

## 7 Further Analysis

In this section we further examine the effects of architectural choices in $\mu$GCD and other category discovery methods. Section 7.1 seeks to understand the performance gains yielded by *cosine classifiers*, and section 7.2 visualizes the feature spaces learned by different GCD methods.

### 7.1 Understanding cosine classifiers in category discovery

Cosine classifiers with entropy regularization have been widely adopted in recognition settings with limited supervision [14, 25], including in category discovery [19, 43]. In fig. 3, we provide justifications for this by inspecting the norms of the learned vectors in the final classification layer.

Specifically, consider a classifier (without a bias term) as $g = \mathbf{W} \in \mathbb{R}^{d \times k}$, containing $k$ vectors of $d$ dimension, one for each output category. In fig. 3, we plot the magnitude of each of these vectors trained with different constraints on CUB [64] (one of the datasets in the SSB [20]). Note that the classifier is constructed such that the first 100 vectors correspond to the 'Old' classes, and are trained with ground truth labels. In our full method, with normalized classifiers, the norm of all vectors is enforced to be the unit norm (blue dashed line). If we remove this constraint (solid orange line), we can see that the norms of vectors which are *not* supervised by ground truth labels (indices 101-200) fall substantially. Then, if we further remove the entropy regularization term (solid green line), the magnitudes of the 'Old' class vectors (indices 1-200) increases dramatically.

This becomes an issue at inference time, with per-class logits computed as:

$$l_m = \langle \mathbf{w}_m, \Phi \rangle = |\mathbf{w}_m||\Phi| \cos(\alpha) \qquad \forall m \in \{1...k\}$$

with the class prediction returned as $\arg \max l_m$. In other words, we show that without appropriate regularisation, our GCD models trivially reduce the weight norm of 'New' class vectors ($|\mathbf{w}_m| \quad \forall m > 100$), leaving all images to be assigned to one of the 'Old' classes. The effects of this are visualized in the right panel of fig. 3, which plots the histogram of class predictions for an unregularized GCD classifier. We can see that exactly zero examples are predicted to 'New' classes.

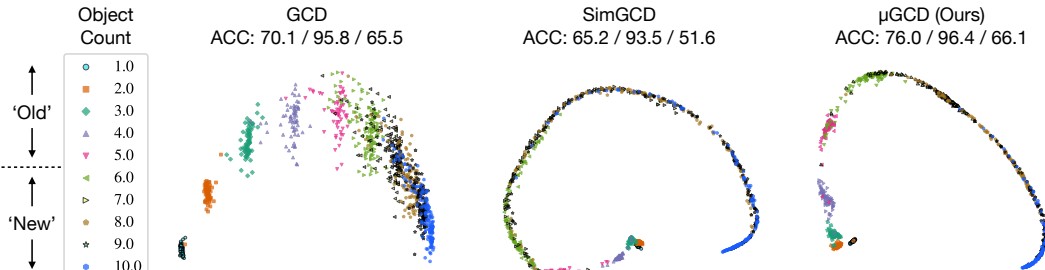

Figure 4: **PCA [71] of features** from the GCD baseline [7], SimGCD [19] and $\mu$GCD on the *count* split of Clevr-4. While the baseline learns eliptical clusters for each category, SimGCD and $\mu$GCD project images onto a one-dimensional object in feature space, which be considered as a 'semantic axis' along which the category changes. Clustering accuracy is reported for 'All'/'Old'/'New' classes.

We further highlight that this effect is obfuscated by the evaluation process, which reports non-zero accuracies for 'New' classes through the Hungarian assignment operation.

## 7.2 PCA Visualization.

Finally, we perform analysis on the *count* split of Clevr-4. Uniquely amongst the four taxonomies, the *count* categories have a clear order. In fig. 4, we plot the first two principal components [71] of the normalized features of the GCD baseline [7], SimGCD [19] and $\mu$GCD. It is clear that all feature spaces learn a clear 'number sense' [72] with image features placed in order of increasing object count. Strikingly, this sense of numerosity is present even beyond the supervised categories (count greater than 5) as a byproduct of a simple recognition task. Furthermore, while the baseline learns elliptical clusters for each category, SimGCD and $\mu$GCD project all images onto a one-dimensional object in feature space. This object can could be considered as a 'semantic axis': a low-dimensional manifold in feature space, $\mathbb{R} \in \mathbb{R}^d$, along which the category label changes.

## 8 Conclusion and final remarks

**Limitations and broader impacts.** In this paper we presented a synthetic dataset for category discovery. While synthetic data allows precise manipulation of the images, and hence more controlled study of the GCD problem, findings on synthetic data do not always transfer directly to real-world images. For instance, the failure case of $\mu$GCD on the *shape* split of Clevr-4 (see section 4.3) occurs due to some classification vectors being unused (see section 7.1). This ceases to be an issue on real-world data with hundreds of categories (see table 5). In appendix F, we fully discuss the difficulties in developing a dataset like Clevr-4 with real-world data. Furthermore, while GCD has many real-world applications, like any form of unsupervised or partially-supervised machine learning, it can be unreliable and must be applied with caution in practise.

**Remarks on Clevr-4.** We note that Clevr-4 can find broader applicability in related machine learning fields. As examples, the dataset can be used for disentanglement research (see appendix F) and as a simple probing set for biases in representation learning. For instance, we find in section 4.1 that most of the ImageNet trained models are biased towards *shape* rather than *texture*, which is in contrast to popular findings from Geirhos *et al.* [13]. Furthermore, larger models are often explicitly proposed as 'all-purpose' features for 'any task' [60]; here we find simple tasks (*e.g.*, *color* or *count* recognition) where initialization with such moodels hurts performance compared to training from scratch. Note that practical problems — *e.g.*, vehicle re-identification [73] or crowd counting [74] — may require understanding of such aspects of the image.

**Conclusion.** In this paper we have proposed a new dataset, Clevr-4, and used it to investigate the problem of Generalized Category Discovery (GCD). This included probing the limitations of unsupervised representations for the task, as well as for identifying weaknesses in existing GCD methods. We further leveraged our findings, together with consistent trends in related literature, to propose a simple but performant algorithm, $\mu$GCD. We find that $\mu$GCD not only provides gains on Clevr-4, but further sets a new state-of-the-art o n the standard Semantic Shift Benchmark.

## Acknowledgments and Disclosure of Funding

S. Vaze is supported by a Facebook Scholarship. A. Vedaldi is supported by ERC-CoG UNION 101001212. A. Zisserman is supported by VisualAI EP/T028572/1.

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

## Appendices

The appendices are summarized in the contents table below. We highlight: appendices A.1 and A.2 for details on Clevr-4; appendix B.4 for further analysis of pre-trained representations on Clevr-4; and appendix C.3 for a long-tail evaluation of $\mu$GCD on the Herbarium19 dataset [66].

## A   Clevr-4

### A.1   Clevr-4 generation

We build Clevr-4 using Blender [54], a free 3D rendering software with a Python API. Following the CLEVR dataset [12], our images are constituted of multiple rendered objects in a static scene. Each object is defined by three 'semantic' attributes (*texture*, *shape* and *color*), and is further defined by its *size*, *pose* and *position* in the scene. We consider the first three attributes as 'semantic' as they are categorical variables which can neatly define image 'classes'. Meanwhile, we designate the *size*, *pose* and *position* attributes as 'nuisance' factors which are not related to the image category.

**CLEVR Limitations.** CLEVR is first limited – for the purposes of category discovery – as it has only two textures ('rubber' and 'metal') and three shapes ('cube', 'sphere' and 'cylinder'). For category discovery, we wish to have *more* categories, both to increase the difficulty of the task, and to ensure a sufficient number of classes in the 'Old' and 'New' subsets. Furthermore, we wish to have the *same* number of categories in each split; otherwise, in principal, an unsupervised algorithm may be able to distinguish the taxonomy simply from the number of categories present.

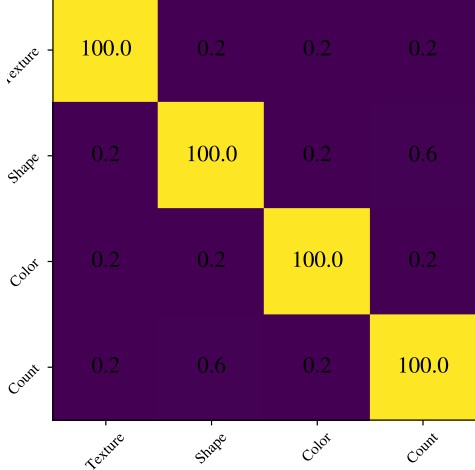

Figure 5: **Normalized mutual information** between the four taxonomies in Clevr-4. All taxonomies have roughly no mutual information between them (they are statistically independent).

**Expanding the taxonomies.** To increase the number of categories in each taxonomy, we introduce new textures, colours and shapes to the dataset, resulting in 10 categories for each taxonomy. We create most of the *8 new textures* by wrapping a black-and-white JPEG around the surface of the object, each of which have their own design (*e.g.*, 'chessboard' or 'circles'). Given an 'alpha' for the opaqueness of this wrapping, these textures can be distinguished independently of the underlying color. We further leverage pre-fabricated meshes packaged with Blender to introduce *7 new shapes* to the dataset, along with *2 new colors* for the objects. The new shapes and colors were selected to be clearly distinguishable from each other. Full definitions of the taxonomies are given in appendix A.2.

**Image sampling process.** For a given image, we first independently sample object *texture*, *shape* and *color*. We then randomly sample how many objects should be in the image (*i.e.*, object *count*) and place this many objects in the scene. Each object has its own randomly sampled size (which is taken to be one of three discrete values), position and relative pose. Thus, differently to CLEVR, all objects in the image have the same *texture*, *shape* and *color*. This allows these three attributes, together with *count*, to define independent taxonomies within the data.

## A.2 Clevr-4 details

We describe the categories in each of the four taxonomies in Clevr-4 below. All taxonomies have 10 categories, five of which are used in the labeled set and shown in bold. Image exemplars of all categories are given in figs. 7 and 8.

- Texture: **rubber**, **metal**, **checkered**, **emojis**, **wave**, `brick`, `star`, `circles`, `zigzag`, `chessboard`
- Shape: **cube**, **sphere**, **monkey**, **cone**, **torus**, `star`, `teapot`, `diamond`, `gear`, `cylinder`
- Color: **gray**, **red**, **blue**, **green**, **brown**, `purple`, `cyan`, `yellow`, `pink`, `orange`
- Count: **1**, **2**, **3**, **4**, **5**, 6, 7, 8, 9, 10

fig. 6 plots the frequency of all categories in the taxonomies, while fig. 5 shows the mutual information between the four taxonomies. We find that all taxonomies, except for *shape*, are roughly balanced, and the four taxonomies have approximately no mutual information between them – realizing our desire of them being *statistically independent*.

## A.3 Clevr-4 examples

We give examples of each of the four taxonomies in Clevr-4 in figs. 7 and 8.

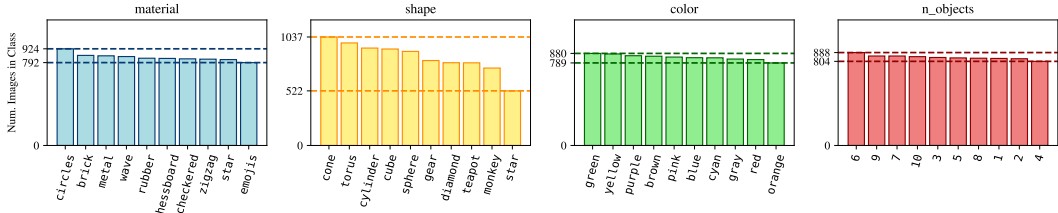

Figure 6: **Category frequency plots** for each taxonomy in Clevr-4. All taxonomies are roughly balanced, except for *shape*. *shape* shows minor imbalance due to greater difficulty in placing many objects of some shapes (*e.g.*, 'star' and 'monkey') in scenes.

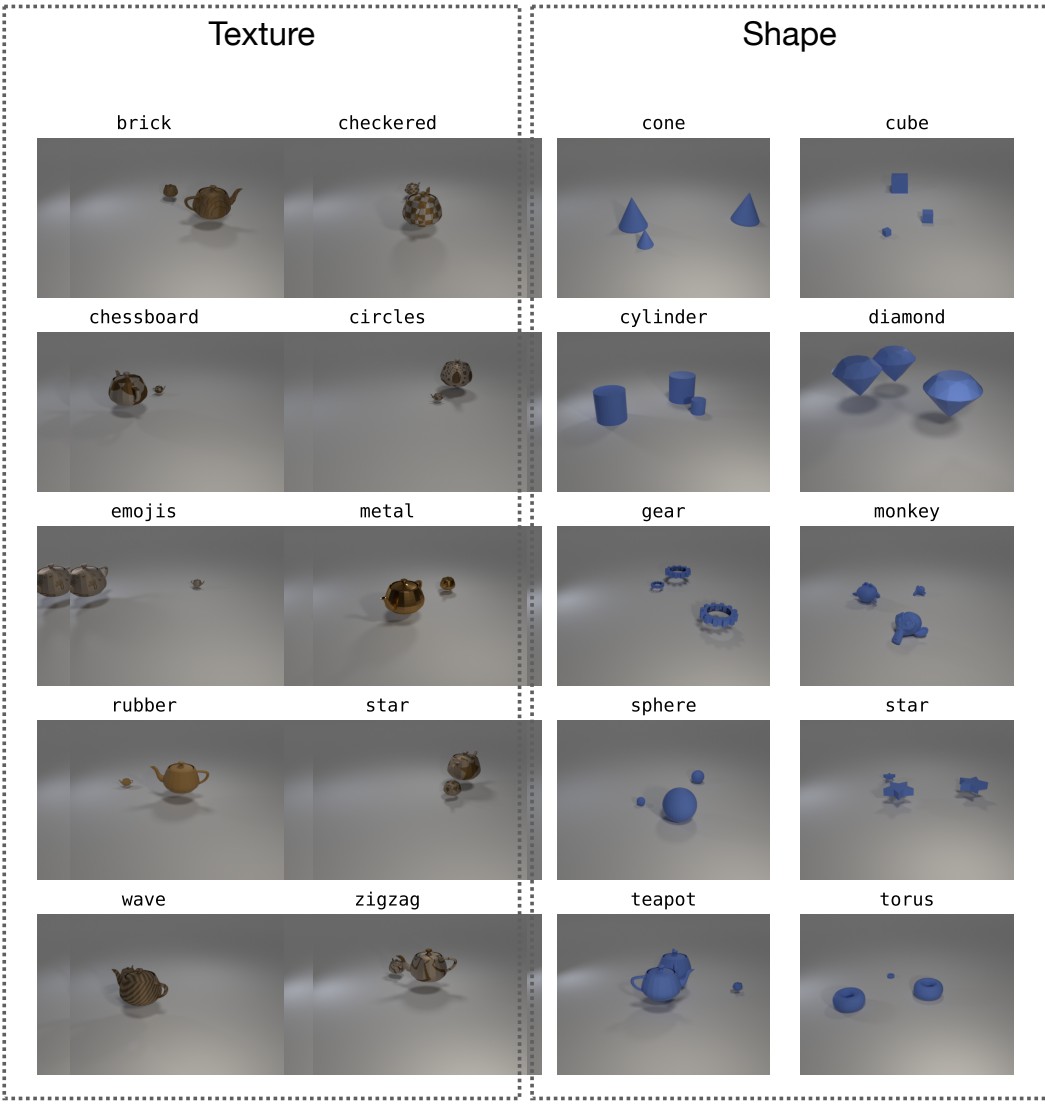

Figure 7: **Examples of each category** from the *texture* and *shape* taxonomies of Clevr-4.

# B  Analysis of results

## B.1  Clevr-4 error bars

We show results for the GCD baseline [7], the current state-of-the-art SimGCD [19] and our method, $\mu$GCD, in fig. 9. The results are shown for five random seeds for each method, and plotted with the

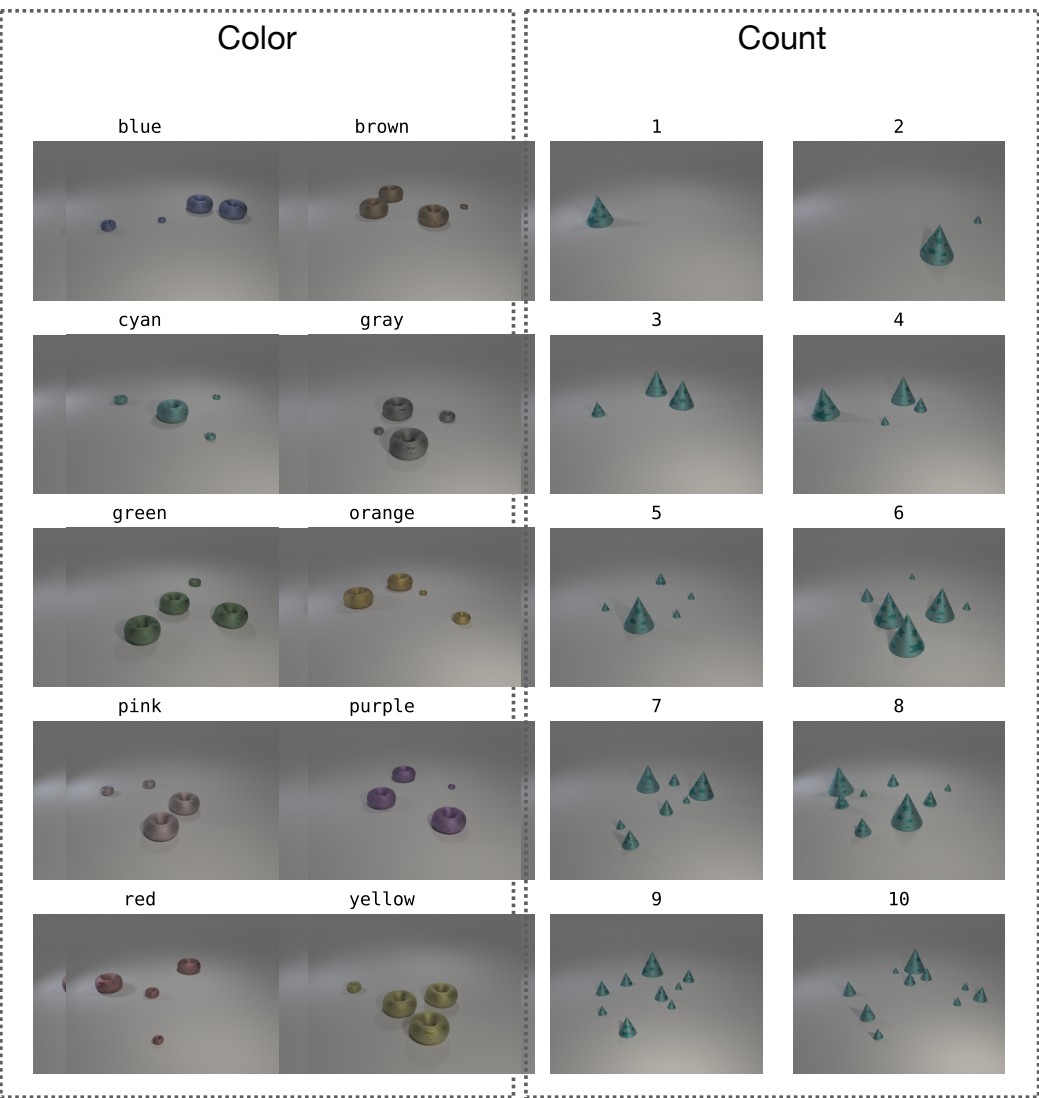

Figure 8: **Examples of each category** from the *color* and *count* taxonomies of Clevr-4.

standard `matplotlib` boxplot function, which identifies outliers in colored circles. We also plot the *median* performance of our method on each taxonomy in dashed lines.

Broadly speaking, the takeaways are the same as the results from Table 4 of the main paper. However, while the *mean* performance of our method is worse than SimGCD on the *shape* split, we can see here that the median performance of $\mu$GCD is *within bounds, or significantly better*, than the compared methods on *all taxonomies*.

## B.2 *shape* failure case

Overall, we find our proposed $\mu$GCD outperforms prior state-of-the-art methods on three of the four Clevr-4 splits (as well as on the Semantic Shift Benchmark [20]). We further show in appendix B.1 that, when accounting for outliers in the five random seeds, our method is also roughly equivalent to the SimGCD [19] state-of-the-art on the *shape* split of Clevr-4.

Nonetheless, we generally find that our method is less stable on the *shape* split of Clevr-4 than on other taxonomies and datasets. We provide some intuitions for this by visualizing the representations and predictions of our method in fig. 10.

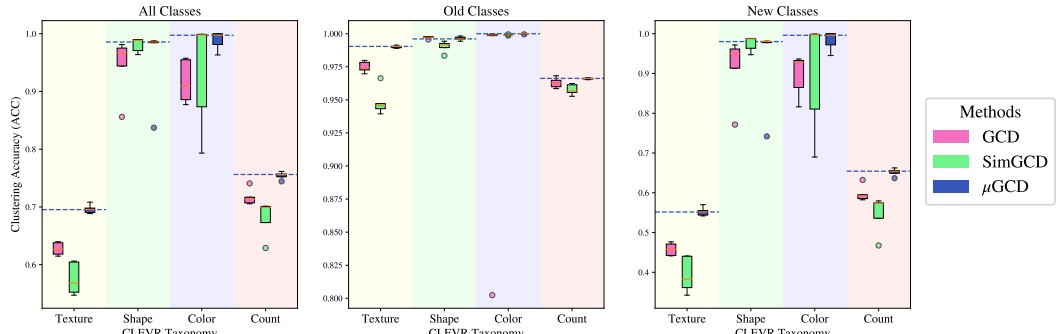

Figure 9: **Box plots of results on Clevr-4.** We show results for the GCD baseline [7], the current state-of-the-art SimGCD [19] and our method, $\mu$GCD. We plot results for five random seeds for the four taxonomies, with outliers shown as colored circles. We also plot the median performance of our method on each taxonomy in dashed lines. On all taxonomies, $\mu$GCD is within bounds, or significantly better, than the compared methods.

**Preliminaries:** In fig. 10, we plot TSNE projections [75] of the feature spaces of two versions of our model, as well as the histograms of the models' predictions on the *shape* split. Along with the models' image representations (colored scatter points), we also plot the class vectors of the cosine classifiers (colored stars). On the left, we show our trained model when we randomly initilize the cosine classifier, while on the right we initialize the class vectors in the classifier with $k$-means centroids. We derive these centroids by running standard $k$-means on the image embeddings of the backbone, which is pre-trained with the GCD-style representation learning step (see appendix D).

**Observations:** In the plot on the **left**, we find that though the feature space is very well separated (there is little overlap between clusters of different categories), the performance of the classifier is still only around 90%. The histogram of model predictions demonstrates that this is due to no images being assigned to the 'star' category – this vector in the classifier is completely unused. Instead, too many instances are assigned to 'gear'. In the TSNE plot, we can see that the 'gear' class vector is between clusters for both 'gear' and 'star' images, while the 'star' vector is pushed far away from both. We suggest that this is due to the optimization falling into a local optimum early on in training, as a result of the feature-space initialization already being so strong.

On the **right**, we find we can largely alleviate this problem by initializing the classification head carefully – with $k$-means centroids from the pre-trained backbone. We see that the problem is nearly perfectly solved, and the histogram of predictions reflects the true class distribution of the labels.

**Takeaway:** We find that when the initialization of the model's backbone – from the GCD-style representation learning step, see appendix D – is already very strong, random initialization of the classification head in $\mu$GCD can result in local optima in the model's optimization process. This can be alleviated by initializing the classification head carefully with $k$-means centroids – resulting in almost perfect performance – but the issue can persist with some random seeds (see appendix B.1).

### B.3    Semi-supervised $k$-means with pre-trained backbones

In fig. 11, we probe the effect of running semi-supervised $k$-means [7] on top of different pre-trained backbones. This is a simple mechanism by which models can leverage the information from the 'Old' class labels. We find that while this improves clustering performance on some taxonomies, it is insufficient to overcome the biases learned during the models' pretraining, corroborating our findings from table 2 of the main paper.

### B.4    Clustering with sub-spaces of pre-trained features

In table 2 and fig. 11, we demonstrate that all pre-trained models have a clear bias towards one of the Clevr-4 taxonomies. Specifically, we find that clustering in pre-trained feature spaces preferentially aligns with a single attribute (e.g *shape* or *color*).

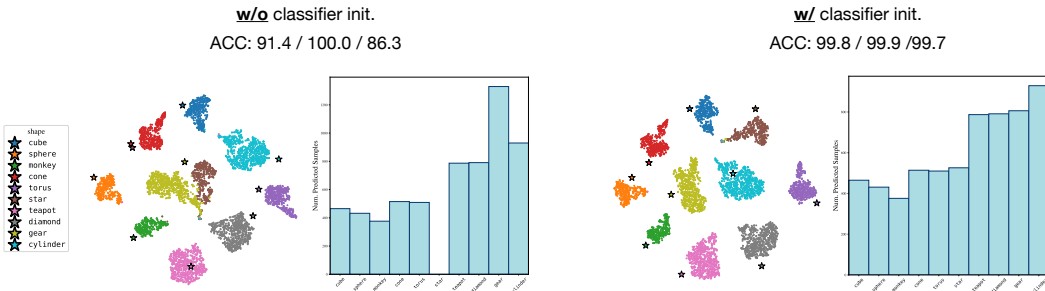

Figure 10: **Analysis of the *shape* failure mode**, showing TSNE plots [75] and prediction histograms for two models, trained *without* (left) and *with* (right) initialization of the classification head with $k$-means centroids. **Left:** When the backbone initialization (from the GCD representation learning step [7]) is already very strong, the classification head gets stuck in a local optimum, with one class vector unused. **Right:** We find we can alleviate this by initializing the class vectors with $k$-means centroids, almost perfectly solving the problem, but the issue can persist with some random seeds.

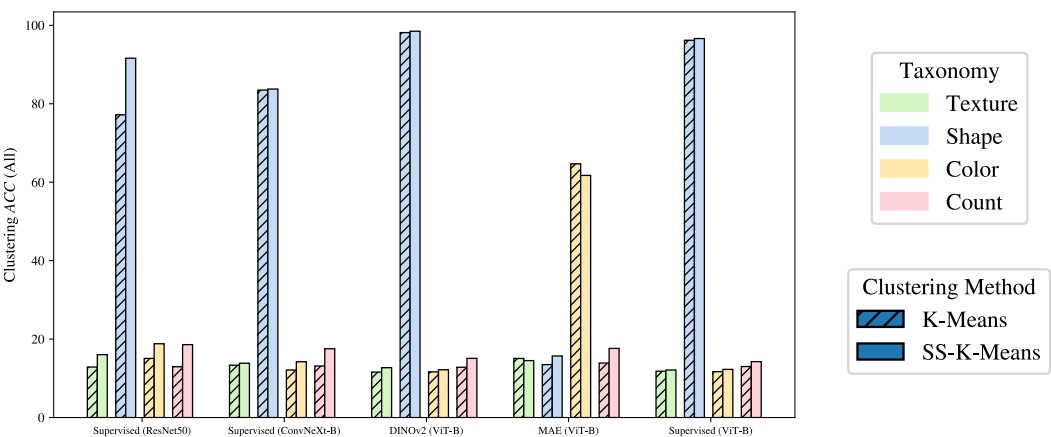

Figure 11: **Effect of semi-supervised $k$-means on representative pre-trained backbones.** We find semi-supervised $k$-means is insufficient to overcome the bias learned during pre-training.

Here, we investigate whether these clusters have any sub-structures. To do this, we perform PCA analysis on features extracted with two backbones: DINOv2 [60] and MAE [31]. Intuitively, we wish to probe whether the omission of dominant features from the backbones (e.g the *shape* direction with DINOv2 features) allows $k$-means clustering to identify other taxonomies. Specifically, we: (i) extract features for all images using a given backbone, $\mathbf{X} \in \mathbb{R}^{N \times D}$; (ii) identify the principal components of the features, sorted by their component scores, $\mathbf{W} \in \mathbb{R}^{D \times D}$; (iii) re-project the features onto the components, omitting those with the $p$ highest scores, $\hat{\mathbf{X}} = (\mathbf{X} - \mu) \cdot \mathbf{W}[:, p :]$; (iv) cluster the resulting features, $\hat{\mathbf{X}} \in \mathbb{R}^{N \times D - p}$, with $k$-means. Here, $\mu$ is the average of the features $\mathbf{X}$, and the results are shown in figs. 12 and 13.

Overall, we find that by removing the dominant features from the backbones, performance on other taxonomies can be improved (at the expense of performance on the 'dominant' taxonomy). The effect is particularly striking with MAE, where we see an almost seven-fold increase in *shape* performance after the the three most dominant principal components are removed.

This aligns with the reported performance characteristics of DINOv2 and MAE. The object-centric recognition datasets on which these models are evaluated benefit from *shape*-biased representations (see section 6). We find here that both MAE and DINOv2 encode shape information, but that more work is required to extract this from MAE features. This is reflected by the strong *linear probe* and *kNN* performance of DINOv2, while MAE requires *full fine-tuning* to achieve optimal performance.

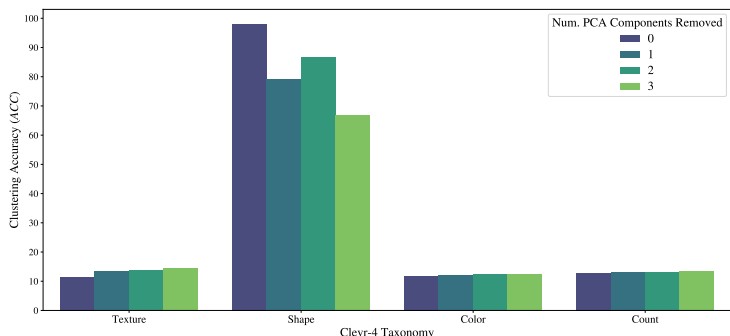

Figure 12: **Re-clustering DINOv2 [60] features after removing dominant principal components.**

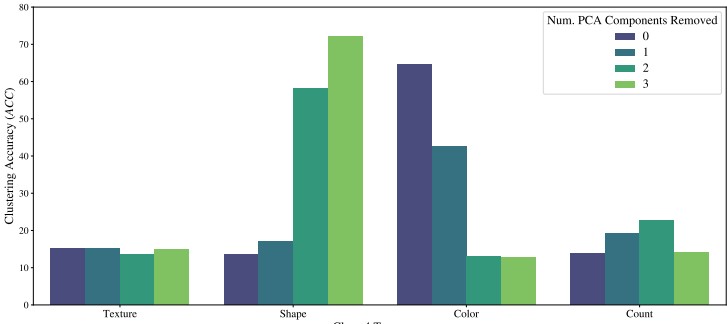

Figure 13: **Re-clustering MAE [31] features after removing dominant principal components.**

Finally, we note that decoding the desired information from pre-trained features is not always trivial, and we demonstrate in section 4 that even in the (partially) supervised, fine-tuning setting in GCD, both of these backbones underperform a randomly initialized ResNet18 on the *count* taxonomy.

### B.5 Design of data augmentation and mis-aligned augmentations

In our SSB experiments, the teacher is passed a weaker augmentation, comprising only `RandomCrop` and `RandomHorizontalFlip`. We find this stabilizes the pseudo-labels produced by the teacher. However, the self-supervised literature consistently finds that strong augmentations are beneficial for representation learning [9, 14, 21]. As such, we experiment with gradually increasing the strength of the augmentation passed to the student model in table 7.

Specifically, we experiment along two axes: the strength of the base augmentation ('Strong Base Aug' column); and how aggressive the cropping augmentation is ('Aggressive Crop' column). To make the base augmentation stronger, we add Solarization and Gaussian blurring [21]. For cropping, we experiment with a light `RandomResizeCrop` (cropping within a range of 0.9 and 1.0) and a more aggressive variant (within a range of 0.3 and 1.0). Overall, we find that an aggressive cropping strategy, as well as a strong base augmentation, is critical for strong performance. We generally found weaker variants to overfit. Though they also have lower peak clustering accuracy, the accuracy falls sharply later in training without the regularization from strong augmentation.

Table 7: **Design of student augmentation.**

| Aggressive Crop | Strong Base Aug | CUB | | |
|:---:|:---:|:---:|:---:|:---:|
| | | All | Old | New |
| ✗ | ✗ | 38.6 | 54.6 | 30.6 |
| ✗ | ✓ | 41.6 | 58.8 | 33.0 |
| ✓ | ✗ | 52.7 | **69.4** | 44.7 |
| ✓ | ✓ | **65.7** | 68.0 | **64.6** |

Table 8: **Effect of mis-aligned augmentations on the GCD Baseline.**

|                        | Color | Count |
|------------------------|-------|-------|
| Aligned Augmentation   | **84.5** | **65.2** |
| Misaligned Augmentation | 26.1  | 46.6  |

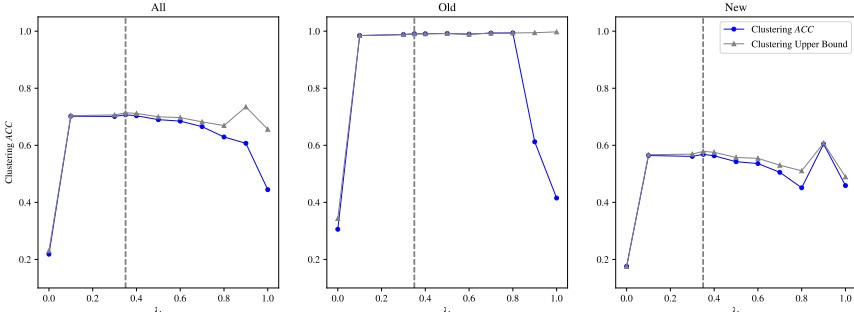

Figure 14: **Effect of hyper-parameter, $\lambda_1$.** We investigate the effect of $\lambda_1$ (which balances the supervised and un-supervised losses), training $\mu$GCD models on the *texture* split.

**Mis-aligned augmentations on Clevr-4** A benefit of Clevr-4 is that each taxonomy has a simple semantic axes. As such, we are able to conduct controlled experiments on the effect of targeting these axes with different data augmentations. Specifically, in table 8, we demonstrate the effect of having 'misaligned' augmentations on two splits of Clevr-4. We train the GCD baseline with `ColorJitter` on the *color* split and `CutOut` for the *count* split. The augmentations destroy semantic information for the respective taxonomies, resulting in substantial degradation of performance. The 'aligned' augmentations are light cropping and flipping for *color*, and light rotation for *count*. The results highlight the importance of data augmentation in injecting inductive biases into deep representations.

## B.6 Effect of $\lambda_1$

In fig. 14, we investigate the effect of the hyper-parameter $\lambda_1$, which controls the tradeoff between the supervised and unsupervised losses in $\mu$GCD. We find that with $0.1 <= \lambda_1 <= 0.4$, the 'All' clustering accuracy is robust, while at $\lambda_1 = 0$ (only unsupervised loss) and $\lambda_1 = 1$ (only supervised loss), performance degrades. We note that the Hungarian assignment in evaluation results in imperfect 'Old' performance even at $\lambda_1$ close to 1 (more weight on the supervised loss). As such, we also show an 'Upper Bound' ('cheating') clustering performance in gray, which allows re-use of clusters in the 'Old' and 'New' accuracy computation.

## C Additional Experiments

### C.1 Results with estimated number of classes

In the main paper, we followed standard practise in category discovery [7, 19, 43, 46, 50, 52] and *assumed knowledge* of the number of categories in the dataset, $k$. Here, we provide experiments when this assumption is removed. Specifically, we train our model using an estimated number of categories in the dataset, where the number of categories is predicted using an off-the-shelf method from [7]. We use estimates of $k = 231$ for CUB and $k = 230$ for Stanford Cars, while these datasets have a ground truth number of $k = 200$ and $k = 196$ classes respectively.

We compare against figures from SimGCD [19] as well as the GCD baseline [7]. As expected, we find our method performs worse on these datasets when an estimated number of categories is used, though we note that the performance of SimGCD [19] improves somewhat on CUB, and the gap between our methods is reduced on this dataset. Nonetheless, the proposed $\mu$GCD still performs marginally better on CUB, and further outperforms the SoTA by nearly 7% on Stanford Cars in this setting.

Table 9: **Results on the SSB with estimated number of categories**. We use the method from [7] to estimate the number of categories as $k = 231$ for CUB, and $k = 230$ for Stanford Cars. We run our method with this many vectors in the classification head, comparing against baselines evaluated with the same estimates of $k$. Results from baselines are reported from [19].

| | Pre-training | CUB | | | Stanford Cars | | | Average |
| --- | --- | --- | --- | --- | --- | --- | --- | --- |
| | | All | Old | New | All | Old | New | All |
| GCD [7] | DINO [14] | 47.1 | 55.1 | 44.8 | 35.0 | 56.0 | 24.8 | 41.1 |
| SimGCD [19] | DINO [14] | 61.5 | **66.4** | 59.1 | 49.1 | 65.1 | 41.3 | 55.3 |
| $\mu$GCD (Ours) | DINO [14] | **62.0** | 60.3 | **62.8** | **56.3** | 66.8 | 51.1 | **59.2** |

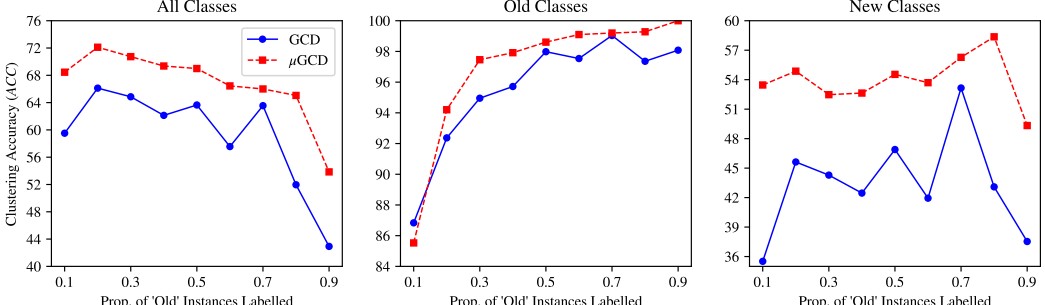

Figure 15: **Results when varying the proportion of 'Old' category images reserved reserved for** $\mathcal{D}_{\mathcal{L}}$. We find our $\mu$GCD method substantially outperforms the GCD baseline [7] across all settings.

## C.2    Results with varying proportion of labelled examples

In the main paper, we follow standard practise in the GCD setting [7, 19, 50, 52] and sample a fixed proportion of images, $p = 0.5$, from the labelled categories and use them in the labeled set, $\mathcal{D}_{\mathcal{L}}$. Here, we experiment with our method if this proportion changes, showing results in fig. 15. We find our proposed $\mu$GCD substantially outperforms the GCD baseline [7] across all tested values of $p$.

## C.3    Results on Herbarium19

We evaluate our method on the Herbarium19 dataset [66]. We use the 'Old'/'New' class splits from [7] which are randomly sampled rather than being curated as they are in the SSB. Nonetheless, the dataset is highly challenging, being long-tailed and containing 683 classes in total. 341 of these classes are reserved as 'Old', and the dataset contains a total of $34K$ images. It further contains a clear taxonomy (herbarium species), making it a suitable evaluation for GCD. We compare $\mu$GCD against prior work in table 10, again finding that we set a new state-of-the-art.

# D    Description of baselines and $\mu$GCD algorithms

In this section we provide step-by-step outlines of: the GCD baseline [7]; the SimGCD [19] baseline; and our method, $\mu$GCD. Full motivation of the design decisions in $\mu$GCD can be found in section 4.

**Task definition and notation:** Given a dataset with labelled ($\mathcal{D}_{\mathcal{L}}$) and unlabelled ($\mathcal{D}_{\mathcal{U}}$) subsets, a model must classify all images in $\mathcal{D}_{\mathcal{U}}$ into one of $k$ possible categories. $\mathcal{D}_{\mathcal{L}}$ contains only a subset of the categories in $\mathcal{D}_{\mathcal{U}}$, and prior knowledge of $k$ is assumed. During training, batches ($\mathcal{B}$), are sampled with both labelled images ($\mathcal{B}_{\mathcal{L}} \in \mathcal{D}_{\mathcal{L}}$) and unlabelled images ($\mathcal{B}_{\mathcal{U}} \in \mathcal{D}_{\mathcal{U}}$). The performance metric is the clustering (classification) accuracy on $\mathcal{D}_{\mathcal{U}}$.

**GCD [7].** Train a backbone, $\Phi$, and perform classification by clustering in its feature space.

**(1)** Train $\Phi$ using an unsupervised InfoNCE loss [23] on all the data, as well as a supervised contrastive loss [61] on the labeled data. Letting $x_i$ and $x'_i$ represent two augmentations of the same image in a batch $\mathcal{B}$, the unsupervised and supervised losses are defined as:

Table 10: **Results on Herbarium19 [66]**, which is a long-tailed recognition dataset.

| | Pre-training | Herbarium19 | | |
| --- | --- | --- | --- | --- |
| | | All | Old | New |
| $k$-means [57] | DINO [14] | 13.0 | 12.2 | 13.4 |
| RankStats+ [46] | DINO [14] | 27.9 | 55.8 | 12.8 |
| UNO+ [43] | DINO [14] | 28.3 | 53.7 | 12.8 |
| GCD [7] | DINO [14] | 35.4 | 51.0 | 27.0 |
| ORCA [8] | DINO [14] | 20.9 | 30.9 | 15.5 |
| OpenCon [52] | DINO [14] | 39.3 | 58.9 | 28.6 |
| PromptCAL [50] | DINO [14] | 37.0 | 52.0 | 28.9 |
| MIB [51] | DINO [14] | 42.3 | 56.1 | 34.8 |
| SimGCD [19] | DINO [14] | 43.3 | 57.9 | 35.3 |
| $\mu$GCD (Ours) | DINO [14] | **45.8** | **61.9** | **37.2** |

$$\mathcal{L}^u_{feat,i} = -\log \frac{\exp\langle \boldsymbol{z}_i, \boldsymbol{z}'_i \rangle / \tau}{\sum_n^{n \neq i} \exp\langle \boldsymbol{z}_i, \boldsymbol{z}_n \rangle / \tau}, \quad \mathcal{L}^s_{feat,i} = -\frac{1}{|\mathcal{N}(i)|} \sum_{q \in \mathcal{N}(i)} \log \frac{\exp\langle \boldsymbol{z}_i, \boldsymbol{z}_q \rangle / \tau}{\sum_n^{n \neq i} \exp\langle \boldsymbol{z}_i, \boldsymbol{z}_n \rangle / \tau}$$

where: $\boldsymbol{z}_i = h \circ \Phi(\boldsymbol{x}_i)$; $h$ is a *projection head*, which is used during training and discarded afterwards; and $\tau$ is a temperature value. $\mathcal{N}(i)$ represents the indices of images in the labeled subset of the batch, $\mathcal{B}_\mathcal{L} \in \mathcal{B}$, which belong to the same category as $\boldsymbol{x}_i$. Given a weighting coefficient, $\lambda_1$, the total contrastive loss on the model's features is given as:

$$\mathcal{L}_{feat} = (1 - \lambda_1) \sum_{i \in \mathcal{B}} \mathcal{L}^u_{feat,i} + \lambda_1 \sum_{i \in \mathcal{B}_\mathcal{L}} \mathcal{L}^s_{feat,i} \qquad (4)$$

(**2**) Perform classification by embedding all images with the trained backbone, $\Phi$, and apply semi-supervised $k$-means (SS-$k$-means) clustering on the entire dataset, $\mathcal{D}_\mathcal{U} \bigcup \mathcal{D}_\mathcal{L}$. SS-$k$-means is identical to unsupervised $k$-means [57] but, at each iteration, instances from $\mathcal{D}_\mathcal{L}$ are always assigned to the 'correct' cluster using their labels, before being used in the centroid update. In this way, the cluster centroid updates for labelled classes are guided by the labels in $\mathcal{D}_\mathcal{L}$.

**SimGCD [19].** Train a backbone representation, $\Phi$, and a linear head, $g$, to classify images amongst the $k$ classes in the dataset, yielding a model $f_\theta = g \circ \Phi$. Train the backbone *jointly* with the feature space loss from eq. (4), and with linear classification losses based on the output of $g$.

(**1**) Generate *pseudo-labels* for an image, $\boldsymbol{x}_i$, as $\boldsymbol{p}_T(\boldsymbol{x}_i) \in [0,1]^k$, in order to train the classifier, $f_\theta$. Infer pseudo-labels on all images in a batch, $\mathcal{B}$, and compute an additional supervised cross-entropy loss on the labelled subset, $\mathcal{B}_\mathcal{L}$.

- Pass two views of an image to the *same model*. Each view generates a soft pseudo-label for the other, for instance as:

$$\boldsymbol{p}_T(\boldsymbol{x}_i) = \text{sg}[\text{softmax}(f_\theta(\boldsymbol{x}'_i); \tau_T)] \qquad (5)$$

Here sg is the stop-grad operator and $\tau_T$ is the pseudo-label temperature.

- Compute model predictions as $\boldsymbol{p}_S(\boldsymbol{x}) = \text{softmax}(f_\theta(\boldsymbol{x}); \tau_S)$ and a standard pseudo-labelling loss [14, 18, 26] (*i.e.* soft cross-entropy loss) as:

$$\mathcal{L}^u_{cls}(\theta; \mathcal{B}) = -\frac{1}{|\mathcal{B}|} \sum_{\boldsymbol{x}_i \in \mathcal{B}} \langle \boldsymbol{p}_T(\boldsymbol{x}_i), \log(\boldsymbol{p}_S(\boldsymbol{x}_i)) \rangle + \langle \boldsymbol{p}_T(\boldsymbol{x}'_i), \log(\boldsymbol{p}_S(\boldsymbol{x}'_i)) \rangle \qquad (6)$$

Temperatures are chosen such that $\tau_T < \tau_S$ to encourage confident pseudo-labels [14].

- Optimize the model, $f_\theta$, *jointly* with: the pseudo-label loss (eq. (6)) and $\mathcal{L}_{feat}$ (see eq. (4)). The model is further trained with: the standard supervised cross-entropy loss on the labelled subset of the batch, $\mathcal{L}^s_{cls}(\theta; \mathcal{B}_\mathcal{L})$; and an entropy regularization term, $\mathcal{L}^r_{cls}(\theta)$:

$$\mathcal{L}^s_{cls}(\theta; \mathcal{B}_\mathcal{L}) = -\frac{1}{|\mathcal{B}_\mathcal{L}|} \sum_{i \in \mathcal{B}_\mathcal{L}} \langle \boldsymbol{y}(\boldsymbol{x}), \log(\boldsymbol{p}_S(\boldsymbol{x})) \rangle, \quad \mathcal{L}^r_{cls}(\theta) = -\langle \bar{\boldsymbol{p}}_S, \log(\bar{\boldsymbol{p}}_S) \rangle, \bar{\boldsymbol{p}}_S = \frac{1}{|\mathcal{B}|} \sum_{\boldsymbol{x} \in \mathcal{B}} \boldsymbol{p}_S(\boldsymbol{x})$$

Here, $\boldsymbol{y}(\boldsymbol{x})$ is a ground-truth label and, given hyper-parameters $\lambda_1$ and $\lambda_2$, the total loss is defined as:
$\mathcal{L}(\theta; \mathcal{B}) = (1 - \lambda_1)(\mathcal{L}^u_{cls}(\theta; \mathcal{B}) + (\mathcal{L}^u_{feat}(\theta; \mathcal{B})) + \lambda_1(\mathcal{L}^s_{cls}(\theta; \mathcal{B}_\mathcal{L}) + \mathcal{L}^s_{feat}(\theta; \mathcal{B}_\mathcal{L})) + \lambda_2 \mathcal{L}^r_{cls}(\theta).$

$\mu$**GCD** (Ours). Train a backbone representation, $\Phi$, and a linear head, $g$, to classify images amongst the $k$ classes in the dataset, yielding a model $f_{\theta_T} = g \circ \Phi$. Train the backbone *first* with the feature space loss from eq. (4), and *then* with linear classification losses based on the output of $g$.

**(1)** Train a backbone $\Phi$ using Step (1) from the GCD baseline algorithm.

**(2)** Append a classifier, $g$, to the backbone and duplicate it to yield two models. One model (a *teacher network*, $f_{\theta_T}$) is used to generate pseudo-labels for a *student network*, $f_{\theta_S}$, as $\boldsymbol{p}_T(\boldsymbol{x}_i) \in [0,1]^k$. Infer pseudo-labels on all images in a batch, $\mathcal{B}$, and compute an additional supervised cross-entropy loss on the labelled subset, $\mathcal{B}_\mathcal{L}$. The student and teacher networks are trained as follows:

- Generate a *strong augmentation* of an image, $\boldsymbol{x}_i$, and a *weak augmentation*, $\boldsymbol{x}'_i$ [49]. Pass the weak augmentation to the *teacher* to generate a pseudo-label and construct a loss:

$$\boldsymbol{p}_T(\boldsymbol{x}_i) = \text{sg}[\text{softmax}(f_{\theta_T}(\boldsymbol{x}'_i); \tau_T)] \quad \mathcal{L}^u_{cls}(\theta_S; \mathcal{B}) = -\frac{1}{|\mathcal{B}|} \sum_{\boldsymbol{x}_i \in \mathcal{B}} \langle \boldsymbol{p}_T(\boldsymbol{x}_i), \log(\boldsymbol{p}_S(\boldsymbol{x}_i)) \rangle \quad (7)$$

- Optimize the student's parameters, $\theta_S$, with respect to: the pseudo-label loss from eq. (7); the supervised loss, $\mathcal{L}^s_{cls}$; and the entropy regularization loss, $\mathcal{L}^r_{cls}$. Formally, the 'student', $f_{\theta_S}$, is optimized for: $\mathcal{L}(\theta_S; \mathcal{B}) = (1 - \lambda_1)\mathcal{L}^u_{cls}(\theta_S; \mathcal{B}) + \lambda_1 \mathcal{L}^s_{cls}(\theta_S; \mathcal{B}_\mathcal{L}) + \lambda_2 \mathcal{L}^r_{cls}(\theta_S).$
- Update the teacher network's parameters with the Exponential Moving Average (EMA) of the student network [17]. Specifically, update the 'teacher' parameters, $\theta_T$, as:

$$\theta_T = \omega(t)\theta_T + (1 - \omega(t))\theta_S$$

where $t$ is the current epoch and $\omega(t)$ is a time-varying decay schedule.

At the end of training, the 'teacher', $f_{\theta_T}$, is used for evaluation.

**Remarks:** We first highlight the different ways in which the labels from $\mathcal{D}_\mathcal{L}$ are used between the three methods. Specifically, the GCD baseline [7] only uses the labels in a feature-space supervised contrastive loss. However, in addition to this, SimGCD [19] and $\mu$GCD *also* use the labels in a standard cross-entropy loss in order to train part of a linear classifier, $g$.

We further note the high level similarity between SimGCD and $\mu$GCD, in that both train parametric classifiers with a pseudo-label loss. While SimGCD uses different views passed to the same model to generate pseudo-labels for each other (similarly to SWaV [9]), $\mu$GCD uses pseudo-labels from a 'teacher' network to train a 'student' (similarly to mean-teachers [17]).

This is in keeping with trends in related fields, which find that there exists a small kernel of methodologies — *e.g.*, mean-teachers [17], cosine classifiers [62], entropy regularization [26] — which are robust across many tasks [14, 22, 25], but that *finding a strong recipe* for a specific problem is critical. We find this to be true in supervised classification [58, 76, 77], self-supervised learning [14, 21], and semi-supervised learning [17, 26, 49]. Our use of mean-teachers to provide classifier pseudo-labels, as well as careful choice of model initialization and data augmentation, yields a performant $\mu$GCD algorithm for category discovery.

# E   Further Implementation Details

When re-implementing prior work, we aim to follow the hyper-parameters of the GCD baseline [7] and SimGCD [19], and use the same settings for our method. We occasionally find that tuned hyper-parameters are beneficial in some settings, which we detail below.

**Learning rates.** We swept learning rates at factors of 10 for all methods and architectures. When training models from scratch (ResNet18 on Clevr-4) or when finetuning a DINO/DINOv2 model [14, 60] on the SSB, we found a learning rate of 0.1 to be optimal. When finetuning an MAE [31] or DINOv2 model on Clevr-4, we found it better to lower the learning rate to 0.01. All learning rates are decayed from their initial value by a factor of $10^{-3}$ throughout training with a cosine schedule.

**Loss hyper-parameters.** For the tradeoff between the unsupervised and supervised components of the losses, $\lambda_1$ is set to 0.35 for all methods. For the entropy regularization, we follow SimGCD and use $\lambda_2 = 1.0$ for FGVC-Aircraft and Stanford Cars, and $\lambda_2 = 2.0$ for all other datasets. We swept to find better settings for this term on Clevr-4, but did not find any setting to consistently improve results. We also train with $L^2$ weight decay, set to $10e^{-4}$ for all models.

**Student and teacher temperatures.** Following [14], we set the temperature of the student and teacher to $\tau_S = 0.1$ and $\tau_T = 0.04$ respectively, for both our method and SimGCD. This gives the teacher 'sharper' (more confident) predictions than the student. We further follow the teacher-temperature warmup schedule from [14], also used in SimGCD, where the teacher temperature is decreased from 0.07 to 0.04 in the first 30 epochs of training. On Herbarium19 [66] (which has many more categories than the other evaluations, see appendix C.3), we use a teacher temperature of $2 \times 10^{-3}$ (warmed up from $3.5 \times 10^{-3}$ over 10 epochs).

**Teacher Momentum Schedule.** In $\mu$GCD, at each iteration, the teacher's parameters are linearly interpolated between the teacher's current parameters and the student's, with the interpolation ('decay' or 'momentum') changing over time following [18], as: $\omega(t) = \omega_T - (1 - \omega_{base})(\cos(\frac{\pi t}{T}) + 1)/2$.

Here $T$ is the total number of epochs and $t$ is the current epoch. We use $\omega_T = 0.999 \approx 1$ and $\omega_{base} = 0.7$. We note for clarity that, though the momentum parameter is dictated by the *epoch* number, the teacher update happens at each *gradient step*.

**Augmentations.** On Clevr-4 we use an augmentation comprising of `RandomHorizontalFlip` and `RandomRotation`. On the SSB [20], we use `RandomHorizontalFlip` and `RandomCrop`. We use these augmentations for all methods, and for $\mu$GCD use these augmentations to pass views to the 'teacher'. An important part of our method on the SSB is to design **strong** augmentations to pass to the student. Our 'strong augmentation' adds aggressive `RandomResizeCrop`, as well as solarization and Gaussian blurring [21] (see appendix B.5 for details). On Clevr-4, due to the relatively simple nature of the images, strong augmentations can destroy the semantic image content; for instance color jitter and aggressive cropping degrade performance on *color* and *count* respectively. We find it helpful to pass Cutout [78] to the teacher on the *color* taxonomy, and *texture* benefits from the strong augmentation defined above.

**Training time.** Following the original implementations, we train all SimGCD [19] and GCD baseline [7] models for 200 epochs, which we find sufficient for the losses (and validation performance) to plateau. For our method, we randomly initialize a classifier on a model which has been trained with the GCD baseline loss, and further finetune for another 100 epochs. On our hardware (either an NVIDIA P40 or M40) we found training to take roughly 15 hours for SSB datasets, and around 4 hours for a Clevr-4 experiment.

**Early stopping.** We note that GCD is a *transductive* setting, or a *clustering* problem, where models are trained (in an unsupervised fashion) on the data used for evaluation, $\mathcal{D}_{\mathcal{U}}$. As such, an important criterion is *which metric* to use to select the best model. SimGCD [19] and the GCD baseline [7] use the performance on a validation set of images from the labeled categories. While this is a reasonable choice for the baseline, we found it can lead to underestimated performance for SimGCD on some datasets. For SimGCD, we instead found it better to simply take the model at the end of training. For $\mu$GCD, we instead propose to choose the model with the minimum *unsupervised loss* on the unlabeled set.

**Other details:** When finetuning pre-trained transformer models – DINO [14], DINOv2 [60] or MAE [31] – we finetune the last transformer block of the model. For Clevr-4, when training a ResNet18, we finetune the whole model. Finally, for the $\mu$GCD failure case of *shape*, we suggest in appendix B.2 that $\mu$GCD can get stuck in local optima if its initialization is already very strong. As such, in this case, we initialize the linear head with $k$-means centroids, reduce the learning rate and teacher temperature to 0.01, and set $\omega_{base}$ to 0.9.

# F  Connections to related work

## F.1  Clevr-4: connections to real-world and disentanglement datasets

**Datasets with different granularities.**  When multiple taxonomies are defined in exisiting datasets, they are most often specified only at different *granularites*, for instance in CIFAR100 [53], FGVC-Aircraft [55] and iNaturalist [79]. While recognition at different granularites is related to our task – and was explored in [80] – the constituent taxonomies are not *statistically independent*, as the Clevr-4 splits are. We note that, given the number of categories in each taxonomy, an unsupervised model could in principal solve the clustering problem at the different granularities.

**CUB-200-2011 [64].**  Fei *et al.* [63] discuss the existence of alternate, but valid, clusterings of images from fine-grained datasets like CUB [64] – *e.g.*, based on pose or background. We note that the CUB 'Birds' dataset presents an opportunity for constructing an interesting dataset for category discovery. Each image in CUB is labelled for presence (or absence) of each of 312 attributes, where these attributes come from different *attribute types*. Each attribute type (*e.g.*, 'bill shape', 'breast color') provides a different taxonomy with respect to which to cluster the data. However, we found these attribute annotations are too noisy to yield meaningful conclusions.

**Disentanglement datasets.**  We suggest that Clevr-4 is also a useful benchmark for disentanglement research [34, 35]. This research field aims to learn models such that the ground-truth data generating factors (*i.e.*, attributes of an object) are encoded in different subspaces of the image representation. The current CLEVR dataset [12] cannot be used easily for this, as its images contain multiple objects, each with different attributes. Instead, in Clevr-4, all objects share the same attributes, allowing each image to be fully parameterized by the object *shape*, *texture*, *color* and *count*. Furthermore, compared to synthetic datasets for disentanglement [38], Clevr-4 contains more categorical taxonomies, as well as more classes within those taxonomies.

Finally, we note that there exist other extensions of the CLEVR dataset [12], such as ClevrTex [81], Super-CLEVR [82] and CLEVR-X [83], which also add new textures and/or categories to the original datasets. However these datasets *cannot* be used for category discovery (or disentanglement) research as, unlike in Clevr-4, they contain scenes with objects of differing attributes. As such, each image cannot be parameterized with respect to the object attributes in a way which gives rise to clear taxonomies.

**Other related fields**  The GCD task and the Clevr-4 dataset are related to a number of other machine learning sub-fields. *Conditional Similarity* research [84–86] aims to learn different embedding functions given different *conditions*. For instance, the GeneCIS benchmark [84] evaluates the ability of models to retrieve different images given a query and different conditioning text prompt. Meanwhile, the *multiple clustering* [87, 88] and *self-supervised learning* [89, 90] fields investigate the how different choices of data augmentation result in different clusterings of the data. The self-supervised field particularly aims to understand why these inductive biases result in different generalization properties [91–93].

We hope that Clevr-4 can be complementary to these works, and provide a test-bed for controlled experimentation of these research questions.

## F.2  $\mu$GCD method.

We note here that the idea of momentum encoders has been widely used in representation learning [18, 24, 50], semi-supervised learning [17, 25], or to update class prototypes in category discovery [52, 94]. We use a mean-teacher model *end-to-end*, for the backbone representation and the classification head. We highlight that, similar to a rich vein of literature in related fields [14, 21, 26, 58, 76, 77], our goal is to find a specific recipe for the GCD task.

