# OpenReview forum: "No Representation Rules Them All in Category Discovery"
_NeurIPS.cc/2023/Conference — NeurIPS 2023 poster_

### Official Review · Reviewer_aYTM · 2023-07-05

**Soundness:** 3 good
**Presentation:** 4 excellent
**Contribution:** 3 good
**Rating:** 7
**Confidence:** 4

**Summary:**

This paper works on generalized category discovery, a setting that requires classifying unlabelled samples according to the taxonomy defined by the labeled set. This work identifies that a shortcut exists in previous benchmarks, and methods overlooking the classification metric defined by the labeled set could still perform well. It thus proposes the CLEVR-4 benchmark, in which the taxonomy/isolation of categories is separated as 'shape', 'texture' or 'color', or 'count'. Based on this benchmark, it evaluates the performance of previous methods in terms of pre-training/representation and pseudo-labeling and proposes $\mu$GCD. The technique shows better performance in both the proposed benchmark and prior benchmarks.

**Strengths:**

*Originality & significance.* The proposed benchmark is absolutely novel and original. It identifies a key issue in current GCD benchmarks: unsupervised classification algorithms are generally specialized for semantics, and the task could be easily suited without relying on the labeled set. The benchmark is in line with a recent trend of aligning machine outputs with human requirements: what is the definition needed by humans? I am happy to see this benchmark and the revisiting of previous works that it enables.

*Quality.* The benchmark is accompanied by careful evaluations of different aspects of previous methods (representation & classification), and analytical figures (fig. 3/6/7) that help understand the source of improvement. Ablation study and error bars are also provided, which is good and makes the contribution sound.

*Clarity* The presentation is natural and fluent, and I did not find issues in delivery.

*Reproducibility.* Technical details are well provided to facilitate reproduction. Hope to see the dataset soon.

**Weaknesses:**

I have to say that my major concerns are anticipated and well discussed in the appendix. There is one small concern on methodology and I list it in the next section.

**Questions:**

One concern on methodology is whether $\mu$GCD could be listed as a major contribution. It is somehow technically incremental to me and is more like SimGCD Plus (momentum teacher & weak-strong aug) rather than a new method. I understand it suits the proposed benchmark better, and I feel okay with the overall contribution of this paper, the only concern is whether could it be listed as a side-by-side contribution with the benchmark.

**Limitations:**

have been discussed

---

> ### Author Rebuttal · Authors · 2023-08-09
>
> We thank the reviewer for their thorough review of our work!
>
> We first clarify that we will publicly release all code and the Clevr-4 dataset upon paper acceptance. Secondly, regarding the phrasing of our method (**“one concern is whether $\mu$GCD could be listed as a major contribution”**), we propose to clarify our methodological contribution as: “we present a novel algorithm for GCD, $\mu$GCD, which extends the SimGCD method with a mean-teacher based approach”. We are happy to answer any further concerns should they arise.

---

> > ### Comment · Reviewer_aYTM · 2023-08-15
> >
> > Thanks to the authors and I confirm my positive recommendation. I have read the responses from other reviewers and I look forward to seeing the final version (e.g., discussions with DMC & CS).

---

> ### Comment · Reviewer_dGiY · 2023-08-16
> **The novelty of the contributed benchmark is not absolute**
>
> To make sure that the community is not guided by the misconception of this review of the paper, I would like to point out that, the novelty of contribution on the clever-4 benchmark is **not absolute**, as similar lines of research have been conducted in the domains like DMC and CS.
> But I agree with the point that the proposed benchmark could help with the current research on GCD.
> Yet limitations like the small number of classes could limit the potential usefulness of the proposed clevr-4 dataset in practice.

---

> > ### Comment · Reviewer_aYTM · 2023-08-17
> >
> > Thanks for pointing out those related works. I was not aware of DMC & CS before and I indeed find them highly related. I agree that taking them into discussion could certainly benefit this paper, which is promised in the authors' reply. And the major contribution of bringing such a trend into the GCD community is still acceptable. I am also aware of the limitations in terms of small scale and remaining synthetic of the proposed dataset, which is also discussed in the paper. Overall, the aforementioned improvements could definitely make it a better work, but from my expertise, it already reaches the bar of NeurIPS.

---

### Official Review · Reviewer_dGiY · 2023-07-06

**Soundness:** 3 good
**Presentation:** 3 good
**Contribution:** 3 good
**Rating:** 5
**Confidence:** 5

**Summary:**

This paper contributes one new dataset and one new method for category discovery. The dataset is designed in a way that each image can be clustered into 4 different clustering based on the attribute of the image (shape, count, texture, and color), this design can be used to reveal the difficulty of unsupervised clustering and the necessity of category discovery methods. The proposed method extends on the SimGCD method by adding an EMA teacher and strong-weak augmentations.

**Strengths:**

1. I really like the discussion of the limitation of unsupervised clustering method on self-supervised representations.
2. The design of the Clevr-4 dataset is interesting, and could be a direction for future works.

**Weaknesses:**

1. I would like to note a few related fields, (1) deep multiple clustering [R1, R2] which deals with unsupervised clustering for different clustering criteria, and (2) conditional similarity [R3, R4]
2. The proposed method seems to be a strighforward extension of the SimGCD method, I think it is not significant enough to be a claimed contribution.
3. I think the discussion on the experiment of previous method on Clevr-4 is not comprehensive, as mentioned in line 262-269, augmentations seem to play an important role for category discovery, then I think the discussion on Clevr-4 should contain some experiments of using different augmentation for differe splits.


[R1] AugDMC: Data Augmentation Guided Deep Multiple Clustering, https://arxiv.org/abs/2306.13023

[R2] A Diversified Attention Model for Interpretable Multiple Clusterings, TKDE2022

[R3] Learning Similarity Conditions Without Explicit Supervision, CVPR 2019

[R4] Towards Latent Attribute Discovery From Triplet Similarities, ICCV 2019


**Questions:**

1. I am interested in how would applying semi-supervised k-means algorithm on pretrained models in tab 2 performs, because using the semi-supervised k-means is IMO the easiest way to inject certain clustering criteria to the model.
2. Since the proposed method is similar to SimGCD, then how about applying the strong augmentation to SimGCD or GCD and compare with that performance? It seems that the strong augmentation plays an important role for the performance as shown in line 2 of tab 6.
3. I am also interested in the performance of tuning the augmentations used by SimGCD and GCD method on Clevr-4. As `misaligned augmentations degrade performance`, yet there are no experiment supporting this claim.

**Limitations:**

I think the methodology contribution of this paper is limited, the proposed method seems like a combination of FixMatch and SimGCD.
The proposed Clevr-4 might be of interest to the community, but as I discussed in the weakness and questions section, I think currently the discussion on Clevr-4 is not comprehensive enough.

---

> ### Author Rebuttal · Authors · 2023-08-09
>
> We thank the reviewer for their detailed feedback, and hope to address their concerns as follows:
>
> **“How would applying semi-supervised $k$-means algorithm on pre-trained models in tab 2 perform”**:
>
> We have re-evaluated a representative selection of the backbones from Tab 2 in the paper, in both unsupervised and semi-supervised k-means settings, for all four Clevr-4 taxonomies. The results are shown in the Author Response PDF (Fig R2). Overall, we find that though semi-supervised k-means can sometimes marginally improve performance on some taxonomies, it is *far from sufficient* to overcome the bias from the model’s pre-training. We suggest that this points to the utility of Clevr-4 as a test-bed for GCD, as well as as a probing set for biases in pre-trained representations.
>
> **“The discussion on Clevr-4 should contain some experiments of using different augmentation for different splits”**:
>
> In the Author Response PDF (Tab R1), we have now included experiments of the GCD baseline trained on different taxonomies with different augmentations. Specifically, for the *count* split, we trained the model with ‘CutOut’, which blacks out a large portion of the image and hence hides a number of the objects. On the *color* split, we trained the model with ‘ColorJitter’, which disturbs the colors in the image. As such, these augmentations (though commonly used in the vision literature) are ‘misaligned’ with the respective taxonomies, and hence result in substantial performance degradation. We find that such failure modes, though quite intuitive, are difficult to isolate with existing GCD (or SSL) datasets, thus underlining the use case for Clevr-4.
>
> We thank the reviewer for the suggestion, and we will update our manuscript to include this analysis and discussion.
>
> **“...how about applying the strong augmentation to SimGCD or GCD and compare with that performance”**:
>
> This SimGCD experiment is almost exactly the same as the experiment from L(4) of the ablation in Tab 6. In this case, we train our model with $\omega(t)$ always equal to 0, thereby using the weights from the last iteration as the ‘teacher’. Here, we find that performance degrades by 3%, pointing to the importance of the slowly updated mean-teacher used in $\mu$GCD. We thank the reviewer for raising this point, and we will clarify it in the text.
>
> **“I would like to note a few related fields…[deep multiple clustering (DMC), conditional similarity (CS)]”**:
>
> We thank the reviewer for pointing out these related problems, they are certainly relevant and we will discuss them fully in the updated manuscript. We will also conduct investigations to see if techniques from these fields can be brought into GCD. Similarly, we hope that our introduced Clevr-4 benchmark can also find use in DMC/CS research.
>
> **“The proposed method seems to be a straightforward extension of the SimGCD”**:
>
> We thank the reviewer for the comment and we propose to clarify our methodological contributions as: “we present a novel algorithm for GCD, $\mu$GCD, which extends the SimGCD method with a mean-teacher based approach”.

---

> > ### Comment · Reviewer_dGiY · 2023-08-16
> > **Thanks for the response.**
> >
> > **1. Results of semi-supervised $k$-means:**
> >
> > Thanks for performing the experiments, I think this could be a valuable add-on for the paper.
> >
> > **2. Results on using aligned augmentations:**
> >
> > I think given the performance boost of using an aligned augmentation, this should definitely be mentioned in the main paper.
> > As a reader of the paper, my first thought on the different taxonomy of categories is that augmentations could play a key role here.
> > I would like to also note a few references on how augmentations influence the learned representations [a,b].
> > Adding these discussions would make the paper stronger.
> >
> >
> > I would encourage the author to include these discussions, as they further complete the paper.
> > Given that the response resolves my concern, and these changes to the paper is easy to make, I would remain positive about this paper.
> >
> >
> > [a] Amortised Invariance Learning for Contrastive Self-Supervision, ICLR 2023.
> > [b] Why Do Self-Supervised Models Transfer? Investigating the Impact of Invariance on Downstream Tasks.

---

> > > ### Author Response · Authors · 2023-08-20
> > > **Further author response to reviewer dGiY**
> > >
> > > We thank the reviewer for the response and are glad that we could resolve their concern. Regarding their further raised points:
> > >
> > > **“[SS-K-Means experiments] could be a valuable add on to the paper”** and **“I would encourage authors to include these discussions [on mis-aligned augmentation]”**:
> > >
> > > Yes, we will include both of these sets of experiments, and corresponding discussions, in the updated manuscript. While the augmentation results are described in L268, we agree with the reviewer that the message is strengthened with the results provided.
> > >
> > > We will also include discussion of further literature the reviewer suggests [a,b]. As with the DMC and CS literature, we hope that Clevr-4 can be complementary to these papers, providing a test-bed for controlled experimentation.
> > >
> > > [a] Amortised Invariance Learning for Contrastive Self-Supervision, ICLR 2023
> > >
> > > [b] Why Do Self-Supervised Models Transfer? Investigating the Impact of Invariance on Downstream Tasks.

---

> ### Comment · Reviewer_aYTM · 2023-08-15
> **Kind reminder to reviewer dGiY**
>
> Kind reminder: the "Limitations" section is not intended for the reviewers' opinions (this should be part of "Weaknesses"), and it simply asks whether limitations are discussed in the paper (yes or no). The original hint is as follows:
> > Have the authors adequately addressed the limitations and, if applicable, potential negative societal impact of their work (refer to the checklist guidelines on limitations and broader societal impacts: https://neurips.cc/public/guides/PaperChecklist)? If not, please include constructive suggestions for improvement. Authors should be rewarded rather than punished for being up front about the limitations of their work and any potential negative societal impact.

---

> > ### Comment · Reviewer_dGiY · 2023-08-16
> >
> > The original hint does not say "Limitations section is not intended for the reviewers' opinions". I would like to point out that this is only reviewer aYTM's personal interpretation of the limitations sections. And this interpretation should not be used to argue with other reviewers.
> > The contents I put in the limitation section are indeed my view of the limitation of the paper presented at submission time.

---

> > > ### Comment · Reviewer_aYTM · 2023-08-17
> > >
> > > I may need to clarify that my previous comment is just a matter of formatting and *have nothing to do with the content itself*. In fact, I agree
> > > the limitations pointed out by reviewer dGiY are reasonable and could benefit the quality of this paper (e.g., acknowledging DMC & CS). Thanks for your effort!

---

> > ### Comment · Reviewer_dGiY · 2023-08-16
> > **Reasons of the limitations**
> >
> > **Limitation 1**
> >
> > > I think the methodology contribution of this paper is limited, the proposed method seems like a combination of FixMatch and SimGCD.
> >
> > This limitation refers to the original claim of the paper
> >
> > > We present a novel method for GCD, µGCD, inspired by the mean-teacher algorithm.
> >
> > According to https://neurips.cc/public/guides/PaperChecklist on the first point:
> >
> > >Do the main claims made in the abstract and introduction accurately reflect the paper's contributions and scope?
> >
> > I think this is a limitation because the claim in the paper is not accurate, also as pointed out in reviewer aYTm's review, the proposed muGCD method is not very novel.
> >
> >
> > **Limitation 2**
> >
> > > The proposed Clevr-4 might be of interest to the community, but as I discussed in the weakness and questions section, I think currently the discussion on Clevr-4 is not comprehensive enough.
> >
> > This is based on the claim in the paper:
> >
> > > Clevr-4 contains four independent taxonomies and can be used to better study the category discovery problem.
> >
> > Since the Clevr-4 benchmark is synthetic, and limited in the number of categories per taxonomy, I argue that this claim is also not accurate and is a limitation of the paper.
> > Referring to https://neurips.cc/public/guides/PaperChecklist, the fourth point :
> >
> > > Reflect on the scope of your claims
> >
> > The claim that clevr-4 is helpful for better study the category discovery problem is small in scope (synthetic and small categories).
> > Also from the results of using aligned augmentations, we can see that actually clevr-4 shows that augmentations matters, which is a studied problem in representations learning [a,b].
> >
> > [a] Amortised Invariance Learning for Contrastive Self-Supervision, ICLR 2023.
> >
> > [b] Why Do Self-Supervised Models Transfer? Investigating the Impact of Invariance on Downstream Tasks.

---

### Official Review · Reviewer_s1RT · 2023-07-06

**Soundness:** 3 good
**Presentation:** 2 fair
**Contribution:** 3 good
**Rating:** 6
**Confidence:** 4

**Summary:**

1. This paper tackles the problem of generalized category discovery (GCD) and identifies the drawbacks of existing methods that they are verified only with labels for a single clustering of the data. In such a case, the model may simply perform unsupervised clustering, not correctly using the available labels.
2. A synthetic dataset, named “Clevr-4” is proposed which contains four independent taxonomies, i.e., shape, texture, color or count.
3.The authors demonstrate the limitations of unsupervised clustering in the GCD setting and propose a new method, µGCD, based on mean teachers, which outperforms existing baselines on Clevr-4 and sets a new state-of-the-art on the Semantic Shift Benchmark (SSB).


**Strengths:**

1. The motivation of this paper is clear and interesting. The authors propose that existing GCD methods may simply perform unsupervised clustering based on the natural grouping of the data. The issue is overlooked by existing methods and is indeed worth investigating.
2. A new benchmark is proposed which provides four independent sets of labels for a common dataset, which is valuable to extensively verify GCD methods’ ability to extrapolate the taxonomies specified by the labeled data using different pre-trained feature extractors.

**Weaknesses:**

1. Additional recent methods should be compared:
Pu N, Zhong Z, Sebe N. Dynamic Conceptional Contrastive Learning for Generalized Category Discovery[C]//Proceedings of the IEEE/CVF Conference on Computer Vision and Pattern Recognition. 2023: 7579-7588.
2. Fig.2 is misleading. Why there are three 'student predictions'? Does the arrows between them mean that the three losses act sequentially, rather than all together?
3. Since the proposed method adopts a two-stage training strategy, i.e., the proposed method could be regarded as an additional 100 epochs of training based on GCD, the training cost could become a concern.

**Questions:**

1. Additional ablation study on lambda_1 which is used to balance the supervised and unsupervised loss terms, should be added to verify the hyperparameter sensitivity.

**Limitations:**

The motivation of verifying GCD methods on taxonomies not merely on the semantic labels is reasonable. The experiments and analysis are comprehensive. Limitations of the method would be limited technical novelty. Codes are not provided.

---

> ### Author Rebuttal · Authors · 2023-08-09
>
> We thank the reviewer for their comments, and respond to their concerns as follows:
>
> **“Additional recent methods should be compared…[DCCL @ CVPR 2023]”**
>
> We thank the reviewer for bringing this interesting paper to our attention, we were unaware of DCCL @ CVPR 2023 as CVPR 2023 was held after the NeurIPS submission deadline. We will include its results and discuss it in the related work. However, we note that its performance is lower than SimGCD and $\mu$GCD on the Semantic Shift Benchmark (against which we compare), thus confirming that our claim of ‘SoTA’ on the SSB remains valid, even after this paper is taken into account.
>
> **“Additional ablation study on $\lambda_1$…”**:
>
> We note that we inherit the value of $\lambda_1$ from prior work (the GCD baseline and SimGCD). However, we have now also ablated its effect on $\mu$GCD in the Author Response PDF (Fig R1). Overall, we find that while performance degrades at extreme values of lambda – i.e near 0 (with only the unsupervised loss) or near 1 (with only the supervised loss) –  $\mu$GCD is robust to $\lambda_1$ changes in the range $0.1 <= \lambda_1 <= 0.4$.
>
> We thank the reviewer for raising this point and we will include this analysis in the updated manuscript.
>
> **“...the training cost [of $\mu$GCD] could become a concern”**
>
> We found the training time of all models to be feasible under an academic compute budget, with models being trained on a single NVIDIA M40 or P40 in roughly 4 hours on Clevr-4, and in roughly 15 hours on the SSB datasets (see Appendix L250). Furthermore, we found that the losses of all baseline methods had plateaued after 200 epochs, verifying that the improved performance of $\mu$GCD is not simply due to longer training.
>
> **“Fig.2 is misleading. Why are there three 'student predictions'?”**:
>
> The student model only produces one set of predictions, and we show these three times in the diagram to visualize the three losses (which are summed and optimized jointly). We agree that the figure may be confusing and will update the diagram, and accompanying text, to clarify.

---

> > ### Comment · Reviewer_s1RT · 2023-08-17
> >
> > I would like to thank the authors for their answers to my questions. Overall, I think it is interesting to propose the "taxonomy issue" which is neglected by recent GCD research. Conceptually, the proposed problem is novel; technically, muGCD is less novel. I would like to keep my positive rating.

---

> ### Comment · Reviewer_aYTM · 2023-08-15
> **Kind reminder to reviewer s1RT**
>
> Kind reminder: the "Limitations" section is not intended for the reviewers' opinions (this should be part of "Weaknesses"), and it simply asks whether limitations are discussed in the paper (yes or no). The original hint is as follows:
> > Have the authors adequately addressed the limitations and, if applicable, potential negative societal impact of their work (refer to the checklist guidelines on limitations and broader societal impacts: https://neurips.cc/public/guides/PaperChecklist)? If not, please include constructive suggestions for improvement. Authors should be rewarded rather than punished for being up front about the limitations of their work and any potential negative societal impact.

---

### Official Review · Reviewer_V8PZ · 2023-07-09

**Soundness:** 3 good
**Presentation:** 3 good
**Contribution:** 3 good
**Rating:** 5
**Confidence:** 3

**Summary:**

This paper addresses generalized class discovery (GCD) from a unique perspective. The authors argues that current GCD benchmarks are unable to ascertain whether models are using the available labels to solve the GCD task, or simply solving an unsupervised clustering problem. In light of this, this paper introduces a new dataset where the data can be clustered according to different rules, i.e., object count, shape or texture. This paper also proposes a simple method that effectively solves such a problem.

**Strengths:**

- The motivation of this paper is sound.
- The paper is well-written and easy to follow.
- The contribution is convincing.

**Weaknesses:**

- The introduced dataset is focused on synthetic data. It could be more convincing to evaluate on real-world data.
- The proposed method is simple yet powerful, but it's unclear which mechanism solves the GCD problem when multiple clustering rules are available.

**Questions:**

- The proposed is a simple reinvent of existing techniques that are common in semi- and self-supervised literature. As mentioned above, the question is how does such a simple method solves the proposed GCD problem when different clustering rules are acceptable?

Thank the authors for the rebuttal, which addressed most of my concerns. Please include the modifications as mentioned in the discussion in the final version.

**Limitations:**

The limitations have been carefully discussed in the paper.

---

> ### Author Rebuttal · Authors · 2023-08-09
>
> We are grateful for the reviewer's feedback on on our work. We hope to address their concerns as follows:
>
> **“The introduced dataset is focused on synthetic data. It could be more convincing to evaluate on real-world data”**:
>
> We found it difficult to find an appropriate real dataset which contains sufficiently complete annotations to construct several taxonomies over the same images. The closest dataset we found was CUB-200-2011, which contains annotations for different attribute types (e.g ‘head color’, ‘bill shape’ etc). However, upon investigation, we found the annotations to be too noisy to build a proper benchmark.
>
> We discuss this issue in the limitations (L331) and the difficulties of constructing real-world datasets of this form in Appendix 6.1.
>
> Given the difficulties of finding an appropriate real dataset, we consider that the advantages of a synthetic dataset — which allows precise manipulation of the images and hence a more controlled study of the GCD problem — outweigh the disadvantages.
>
> **“...unclear which mechanism [in muGCD] solves the GCD problem when multiple clustering rules are available”**:
>
> In Fig 3 (left), we find that the GCD baseline method can, to some extent, learn a representation which distinguishes images according to the desired taxonomy. Our intuition is to begin from this feature space, and train a linear head on top with strong pseudo-labels (generated with the mean-teacher approach) which is then specialized for the desired taxonomy. We thank the reviewer for raising this important point, and we will update the manuscript to extend our discussion in L216 - 234.

---

> > ### Comment · Reviewer_dGiY · 2023-08-16
> >
> > The point about how muGCD solves the problem of multiple clustering GCD is great.
> > I also share this concern, and I think the response here is a bit vague.
> > Since we have observed a huge performance gain from changing the augmentations, doesn't that mean the real mechanism of the solving of GCD is in the augmentation the model uses (as augmentations influence the representation space)?

---

> > ### Comment · Reviewer_V8PZ · 2023-08-18
> >
> > I thank the authors for the response. I understand the difficulty in real-world data. I agree with Reviewer dGiY that the current response to the real mechanism is still vague. I would appreciate it if more theoretical/empirical analysis could be provided to strengthen the motivation of this paper. At least, the readers can still benefit from the discussion that which augmentation favors what kind of taxonomy if the true mechanism is indeed the augmentation.

---

> > > ### Author Response · Authors · 2023-08-20
> > >
> > > We understand the concerns of reviewers **V8PZ** and **dGiY**.
> > >
> > > Concretely, there are two sources of external information (aside from the raw images) which we use to learn the taxonomy:
> > >
> > > (1) The ground-truth labels for the ‘Old’ classes
> > >
> > > (2) The augmentations in the contrastive loss
> > >
> > > Both sources are used in some way in the baselines, prompting us to find that the GCD baseline can ‘to some extent’ identify the correct taxonomy (Fig 6 and comment above).
> > >
> > > We suggest that $\mu$GCD outperforms these baselines by: (1) integrating the labels into within a stronger pseudo-labelling framework (see L226); and (2) carefully selecting augmentations for use with a ‘strong/weak’ augmentation strategy (see L262). We note that, while augmentations are important (see L262, results in Tab R1), the design choices in the pseudo-labelling framework are *also important* (see ablations in Tab 6).
> > >
> > > We thank both reviewers for raising this point, and *we will provide further analysis on these factors in the final version*. For instance, we will provide visualizations of the learned feature space (similar to Fig 6), when different design choices are made - e.g when mis-aligned augmentations are used.

---

### Author Rebuttal · Authors · 2023-08-09

**Global response**:

We sincerely thank all reviewers for the time they spent reviewing our manuscript, and for their thoughtful feedback. We are encouraged that the reviewers found: the ideas *‘unique’* and *'overlooked by previous methods'* (V8PZ, s1RT); our proposed dataset *‘interesting’*, *'valuable'* and *'absolutely novel and original'* (dGiY, S1RT, aYTM); and overall our analysis to be *'comprehensive'* and our contributions *'sound'* and *'convincing'* (s1RT, V8PZ, aYTM).

We have provided detailed responses to individual reviewers below, and have provided additional experiments suggested by the reviewers in the Author Response PDF. We also clarify that we plan to publicly release all code and the Clevr-4 dataset.

---

> ### Author Response · Authors · 2023-08-22
> **Global response after discussion period**
>
> After the discussion period, we are encouraged that the reviewers ‘remain positive’ about the paper (dGiY, aYTM, s1RT) and that we were able to ‘resolve the concern’ of the reviewer dGiY. We will incorporate the following to the manuscript:
>
>    * Discussion of related work (**dGiY**) on: deep multiple clustering (DMC); conditional similarity (CS); and the study of augmentations in self-supervised learning (SSL).
>    * Analysis on: the results with mis-aligned augmentations, and with semi-supervised $k$-means (**dGiY**, shown in Tab R1 and Fig R2); effect of the hyper-parameter $\lambda_1$ (**s1RT**, shown in Fig R1); and an exploration of how $\mu$GCD design choices effect how different taxonomies are learned (specifically with further feature space visualizations as in Fig 6, **V8PZ**).
>
> Furthermore, we will **clarify the scope** of our contributions (**dGiY**, **aYTM**). In this work, we have proposed a *new synthetic dataset* for controlled exploration of the category discovery problem, allowing evaluation of *multiple clusterings* of the same images. This is related to (and may find broader applicability in) the DMC, CS and SSL literature (reviewer **dGiY**).
>
> We have also *developed an effective algorithm for GCD*. This method *extends SimGCD* with components from the semi-supervised learning literature (as noted by **aYTM**). We note that, similarly to papers in related fields (e.g SimCLR, UNO, SimGCD), our methodological contribution is to design a *performant recipe* for GCD (Appendix L297), which we find outperforms SoTA on the introduced dataset (+4%) and on the SSB suite (+3%).

---

### Decision · Program_Chairs · 2023-09-21

**Decision:**

Accept (poster)

**Comment:**

The paper offers a novel approach to Generalized Class Discovery (GCD) by introducing a new dataset, Clevr-4, that challenges existing benchmarks. It proposes a simple yet effective method but leaves some questions about its mechanisms. The paper is well-executed, featuring thorough evaluations and clarity. However, concerns exist about the method's originality and a lack of comprehensive experimental discussion. Overall, the paper is a valuable contribution to the field, filling a gap in existing benchmarks.

The reviewers are all satisfied with the authors' response and recommend accept. AC reads the paper (roughly), the review, and the discussion, and recommends accept (poster).